# DISENTANGLEMENT ANALYSIS WITH PARTIAL INFORMATION DECOMPOSITION

**Seiya Tokui**
The University of Tokyo
`tokui@g.ecc.u-tokyo.ac.jp`

**Issei Sato**
The University of Tokyo
`sato@g.ecc.u-tokyo.ac.jp`

## ABSTRACT

We propose a framework to analyze how multivariate representations disentangle ground-truth generative factors. A quantitative analysis of disentanglement has been based on metrics designed to compare how one variable explains each generative factor. Current metrics, however, may fail to detect entanglement that involves more than two variables, e.g., representations that duplicate and rotate generative factors in high dimensional spaces. In this work, we establish a framework to analyze information sharing in a multivariate representation with Partial Information Decomposition and propose a new disentanglement metric. This framework enables us to understand disentanglement in terms of *uniqueness*, *redundancy*, and *synergy*. We develop an experimental protocol to assess how increasingly entangled representations are evaluated with each metric and confirm that the proposed metric correctly responds to entanglement. Through experiments on variational autoencoders, we find that models with similar disentanglement scores have a variety of characteristics in entanglement, for each of which a distinct strategy may be required to obtain a disentangled representation.

## 1 INTRODUCTION

Disentanglement is a guiding principle for designing a learned representation separable into parts that individually capture the underlying factors of variation. The concept is originally concerned as an inductive bias towards obtaining representations aligned with the underlying factors of variation in data (Bengio et al., 2013) and has been applied to controlling otherwise unstructured representations of data from several domains, e.g., images (Karras et al., 2019; Esser et al., 2019), text (Hu et al., 2017), and audio (Hsu et al., 2019) to name just a few.

While the concept is appealing, defining disentanglement is not clear. After Higgins et al. (2017), generative learning methods with regularized total correlation have been proposed (Kim & Mnih, 2018; Chen et al., 2018); however, it is still not clear if independence of latent variables is essential for better disentanglement (Higgins et al., 2018). Furthermore, it is not obvious to measure disentanglement given true generative factors. Towards understanding disentanglement, it is crucial to define *disentanglement metrics*, for which several attempts have been made (Higgins et al., 2017; Kim & Mnih, 2018; Chen et al., 2018; Eastwood & Williams, 2018; Do & Tran, 2020; Zaidi et al., 2020); however, there are still problems to be solved.

Current disentanglement metrics may fail to detect entanglement involving more than two variables. In these metrics, one first measures how each variable explains one generative factor and then compares or contrasts them among variables. With such a procedure, we may overlook multiple variables conveying information of one generative factor. For example, let $\mathbf{z} = (\mathbf{z}_1, \mathbf{z}_2)$ be a representation consisting of two vectors, where each variable in $\mathbf{z}_1$ disentangles a distinct generative factor and $\mathbf{z}_2$ is a rotation of $\mathbf{z}_1$ not axis-aligned with the factors. Since any dimension of $\mathbf{z}_2$ alone may convey little information of one generative factor, these metrics do not detect that multiple variables encode one generative factor redundantly. Although this is a simple example, this kind of information sharing may arise in learned representations as well, if not the variables are linearly correlated.

In this work, we present a disentanglement analysis framework aware of interactions among multiple variables. Our key idea is to decompose the information of representation into entangled and disentangled components using *Partial Information Decomposition* (PID), which is a framework in modern information theory to analyze information sharing among multiple random variables (Williams

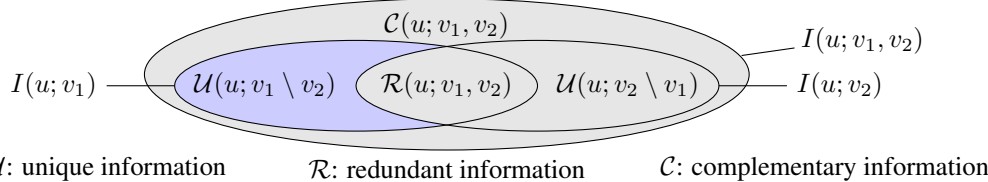

Figure 1: Information diagram of three variable system in PID. Each circle represents mutual information, and each area separated by them represents a decomposed term in PID. When we substitute a generative factor for $u$, a latent variable for $v_1$, and the other latent variables for $v_2$, the unique information $\mathcal{U}(u; v_1 \setminus v_2)$ represents the information of the factor disentangled by the latent variable. See Figure 5 in Appendix for the alternative form similar to the ones we will use in Section 3.

& Beer, 2010). As illustrated in Figure 1, the mutual information $I(u; v_1, v_2) = \mathbb{E}\left[\log \frac{p(u, v_1, v_2)}{p(u)p(v_1, v_2)}\right]$ between a random variable $u$ and a pair of random variables $(v_1, v_2)$ is decomposed into four nonnegative terms: *unique* information $\mathcal{U}(u; v_1 \setminus v_2)$ and $\mathcal{U}(u; v_2 \setminus v_1)$[1], *redundant* information $\mathcal{R}(u; v_1, v_2)$, and *complementary* (or *synergistic*) information $\mathcal{C}(u; v_1, v_2)$. While these partial information terms have no agreed-upon concrete definitions yet (Bertschinger et al., 2014; Finn & Lizier, 2018; Lizier et al., 2018; Finn & Lizier, 2020; Sigtermans, 2020), we can derive universal lower and upper bounds of the partial information terms only with well-defined mutual information terms. We apply the PID framework to representations learned from data by letting $u$ be a generative factor and $v_1, v_2$ be one and the remaining latent variables, respectively. The unique information of a latent variable intuitively corresponds to the amount of disentangled information, while the redundant and complementary information correspond to different types of entangled information. We can quantify disentanglement and multiple types of entanglement through the framework, which enriches our understanding on disentangled representations.

Our contributions are summarized as follows.

- **PID-based disentanglement analysis framework**: We propose a disentanglement analysis framework that captures interactions among multiple variables with PID. With this framework, one can distinguish two different types of entanglement, namely redundancy and synergy, which provide insights on how a representation entangles generative factors.

- **Tractable bounds of partial information terms**: We derive lower and upper bounds of partial information terms. We formulate a disentanglement metric, called UNIBOUND, using the lower bound of unique information. We design *entanglement attacks*, which inject entanglement to a given disentangled representation, and confirm through experiments using them that UNIBOUND effectively captures entanglement involving multiple variables.

- **Detailed analyses of learned representations**: We analyze representations obtained by variational autoencoders (VAEs). We observe that UNIBOUND sometimes disagrees with other metrics, which indicates multi-variable interactions may dominate learned representations. We also observe that different types of entanglement arise in models learned with different methods. This observation provides us an insight that we may require distinct approaches to remove them for disentangled representation learning.

PROBLEM FORMULATION AND NOTATIONS

Let $x$ be a random variable representing a data point, drawn uniformly from a dataset $\mathcal{D}$. Assume that the true generative factors $\mathbf{y} = (y_1, \ldots, y_K)^\top$ are available for each data point; in other words, we can access the subset $\mathcal{D}(\mathbf{y}) \subset \mathcal{D}$ of the data points with any fixed generative factors $\mathbf{y}$. Let $\mathbf{z} = (z_1, \ldots, z_L)^\top$ be a latent representation consisting of $L$ random variables. An inference model is provided as the conditional distribution $p(\mathbf{z}|x)$. Our goal is to evaluate how well the latent variables $\mathbf{z}$ disentangle each generative factor in $\mathbf{y}$. The inference model can integrate out the input as $p(\cdot|\mathbf{y}) = \mathbb{E}_{p(x|\mathbf{y})}[p(\cdot|x)] = \frac{1}{|\mathcal{D}(\mathbf{y})|} \sum_{x \in \mathcal{D}(\mathbf{y})} p(\cdot|x)$; therefore, we only use $\mathbf{y}$ and $\mathbf{z}$ in most of our discussions.

---

[1]Note that this $\setminus$ is not a set difference operator. It is just a common notation used in the PID literature to emphasize the unique information is not symmetric and resembles the set difference as depicted in Fig.1.

We denote the mutual information between random variables $u$ and $v$ by $I(u; v) = \mathbb{E}\left[\log \frac{p(u,v)}{p(u)p(v)}\right]$, the entropy of $u$ by $H(u) = -\mathbb{E}[\log p(u)]$, and the conditional entropy of $u$ given $v$ by $H(u|v) = -\mathbb{E}[\log p(u|v)]$. We denote a vector of zeros by $\mathbf{0}$, a vector of ones by $\mathbf{1}$, and an identity matrix by $\mathbf{I}$. We denote by $\mathcal{N}(\boldsymbol{\mu}, \boldsymbol{\Sigma})$ the Gaussian distribution with mean $\boldsymbol{\mu}$ and covariance $\boldsymbol{\Sigma}$ and denote its density function by $\mathcal{N}(\cdot; \boldsymbol{\mu}, \boldsymbol{\Sigma})$.

## 2 RELATED WORK

The importance of representing data with multiple variables conveying distinct information has been recognized at least since the '80s (Barlow, 1989; Barlow et al., 1989; Schmidhuber, 1992). The *minimum entropy coding* principle (Watanabe, 1981), which aims at representing data by random variables $\mathbf{z}$ with the sum of minimum marginal entropies $\sum_\ell H(z_\ell)$, is found to be useful for unsupervised learning to remove the inherent redundancy in the sensory stimuli. The resulting representation minimizes the total correlation and is called *factorial coding*. Recent advancements in disentangled representation learning based on VAEs (Kingma & Welling, 2014) are guided by the same principle as minimum entropy coding (Kim & Mnih, 2018; Chen et al., 2018; Gao et al., 2019).

Understanding better representations, which is tackled from the coding side as above, is also approached from the generative perspective. It is often expected that data are generated from generative factors through a process that entangles them into high dimensional sensory space (DiCarlo & Cox, 2007). As generative factors are useful as the basis of downstream learning tasks, obtaining disentangled representations from data is a hot topic of representation learning (Bengio et al., 2013). Towards learning disentangled representations, it is arguably important to quantitatively measure disentanglement. In that regard, Higgins et al. (2017) established a standard evaluation procedure using controlled datasets with balanced and fully-annotated ground-truth factors. A variety of metrics have then been proposed on the basis of the procedure. Among them, Higgins et al. (2017) and Kim & Mnih (2018) propose metrics based on the deviation of each latent variable conditioned by a generative factor. In contrast, Mutual Information Gap (MIG) (Chen et al., 2018) and its variants (Do & Tran, 2020; Zaidi et al., 2020) are based on mutual information between a latent variable and a generative factor. We extend the latter direction, considering multi-variable interactions.

Barlow (1989) discussed *redundancy* by comparing the population and the individual variables by their entropies, i.e., total correlation. It is though less trivial to measure redundancy as an information quantity. The PID framework (Williams & Beer, 2010) provides an approach to understanding redundancy among multiple random variables as a constituent of mutual information. The framework provides some desirable relationships between decomposed information terms, while it leaves some degrees of freedom to determine all of them, for which several definitions have been proposed (Williams & Beer, 2010; Bertschinger et al., 2014; Finn & Lizier, 2018; 2020; Sigtermans, 2020).

The PID framework has been applied to machine learning models. For example, Tax et al. (2017) measured the PID terms for restricted Boltzmann machines using the definition of Williams & Beer (2010). Yu et al. (2021) took an alternative route, similar to our approach, where they measured linear combinations of PID terms by corresponding linear combinations of mutual information terms. These work aims at analyzing the learning dynamics of models in supervised settings. In contrast, we use the PID framework for analyzing disentanglement in unsupervised representation learning.

## 3 PARTIAL INFORMATION DECOMPOSITION FOR DISENTANGLEMENT

In this section, we analyze the current metrics and introduce our framework. In Section 3.1, we introduce PID of the system we concern. In Section 3.2, we investigate the current metrics in terms of multi-variable interactions. In Section 3.3 and 3.4, we construct our disentanglement metric with bounds for PID terms. We provide a method of computing the bounds in Section 3.5.

### 3.1 PARTIAL INFORMATION DECOMPOSITION

We tackle the problem of evaluating disentanglement of a latent representation $\mathbf{z}$ relative to the true generative factors $\mathbf{y}$ from an information-theoretic perspective. Let us consider evaluating how one generative factor $y_k$ is captured by the latent representation $\mathbf{z}$. The information of $y_k$ captured by $\mathbf{z}$ is measured using mutual information $I(y_k; \mathbf{z}) = H(\mathbf{z}) - H(\mathbf{z}|y_k)$.

In a desirably disentangled representation, we expect one of the latent variables $z_\ell$ to exclusively capture the information of the factor $y_k$. To evaluate a given representation, we are interested in

understanding how the information is distributed between a latent variable $z_\ell$ and the remaining representation $\mathbf{z}_{\setminus \ell} = (z_{\ell'})_{\ell' \neq \ell}$. This is best described by the PID framework, where the mutual information is decomposed into the following four terms.

$$I(y_k; \mathbf{z}) = \mathcal{R}(y_k; z_\ell, \mathbf{z}_{\setminus \ell}) + \mathcal{U}(y_k; z_\ell \setminus \mathbf{z}_{\setminus \ell}) + \mathcal{U}(y_k; \mathbf{z}_{\setminus \ell} \setminus z_\ell) + \mathcal{C}(y_k; z_\ell, \mathbf{z}_{\setminus \ell}). \qquad (1)$$

Here, the decomposed terms represent the following non-negative quantities.

- *Redundant information* $\mathcal{R}(y_k; z_\ell, \mathbf{z}_{\setminus \ell})$ is the information of $y_k$ held by both $z_\ell$ and $\mathbf{z}_{\setminus \ell}$.

- *Unique information* $\mathcal{U}(y_k; z_\ell \setminus \mathbf{z}_{\setminus \ell})$ is the information of $y_k$ held by $z_\ell$ and not held by $\mathbf{z}_{\setminus \ell}$. The opposite term $\mathcal{U}(y_k; \mathbf{z}_{\setminus \ell} \setminus z_\ell)$ is also defined by exchanging the roles of $z_\ell$ and $\mathbf{z}_{\setminus \ell}$.

- *Complementary information* (or synergistic information) $\mathcal{C}(y_k; z_\ell, \mathbf{z}_{\setminus \ell})$ is the information of $y_k$ held by $\mathbf{z} = (z_\ell, \mathbf{z}_{\setminus \ell})$ that is not held by either $z_\ell$ or $\mathbf{z}_{\setminus \ell}$ alone.

The following identities, combined with Eq.1, partially characterize each term.

$$I(y_k; z_\ell) = \mathcal{R}(y_k; z_\ell, \mathbf{z}_{\setminus \ell}) + \mathcal{U}(y_k; z_\ell \setminus \mathbf{z}_{\setminus \ell}), \quad I(y_k; \mathbf{z}_{\setminus \ell}) = \mathcal{R}(y_k; z_\ell, \mathbf{z}_{\setminus \ell}) + \mathcal{U}(y_k; \mathbf{z}_{\setminus \ell} \setminus z_\ell). \quad (2)$$

The decomposition of this system is illustrated in Figure 2a.

We expect disentangled representations to concentrate the information of $y_k$ to a single latent variable $z_\ell$, and to let the other variables $\mathbf{z}_{\setminus \ell}$ not convey the information in either unique, redundant, or synergistic ways. This is understood in terms of PID as maximizing the unique information $\mathcal{U}(y_k; z_\ell \setminus \mathbf{z}_{\setminus \ell})$ while minimizing the other parts of the decomposition.

The above formulation is incomplete as one degree of freedom remains to determine the four terms with the three equalities. Instead of stepping into searching for suitable definitions of these terms, we build discussions applicable to any such definitions that fulfill the above incomplete requirements[2].

## 3.2 Understanding Disentanglement Metrics from Interaction Perspective

Current disentanglement metrics are typically designed to measure how each latent variable captures a factor and compare it among latent variables, i.e., output a high score when only one latent variable captures the factor well.

For example, the BetaVAE metric (Higgins et al., 2017) is computed by estimating the mean absolute difference (MAD) of two i.i.d. variables following $p(z_\ell | y_k) = \mathbb{E}_{p(x|y_k)} p(z_\ell | x)$ for each $\ell$ by Monte-Carlo sampling and training a linear classifier that predicts $k$ from the noisy estimations of the differences. The FactorVAE metric (Kim & Mnih, 2018) is computed similarly, except that the MAD is replaced with variance (following normalization by population), and a majority-vote classifier is used to eliminate a failure mode of ignoring one of the factors and to avoid depending on hyperparameters. These metrics have the same goal of finding a mapping between $\ell$ and $k$ by comparing the deviation of $z_\ell$ when fixing $y_k$. Since the deviation is computed for each $z_\ell$ separately, these metrics do not count how each latent variable $z_\ell$ interacts with the other variables $\mathbf{z}_{\setminus \ell}$.

Another example is MIG (Chen et al., 2018), which compares mutual information $I(y_k; z_\ell)$ for all $\ell$ and uses the gap between the maximum and the second maximum among them. More precisely, MIG is defined by the following formula.

$$\text{MIG} = \frac{1}{K} \sum_{k=1}^{K} \frac{1}{H(y_k)} \max_\ell \min_{\ell' \neq \ell} (I(y_k; z_\ell) - I(y_k; z_{\ell'})). \qquad (3)$$

Here, dividing each summand by $H(y_k)$ balances the contribution of each factor when they are discrete. The difference in mutual information is rewritten as a difference in unique information as

$$I(y_k; z_\ell) - I(y_k; z_{\ell'}) = \mathcal{U}(y_k; z_\ell \setminus z_{\ell'}) - \mathcal{U}(y_k; z_{\ell'} \setminus z_\ell) \leq \mathcal{U}(y_k; z_\ell \setminus z_{\ell'}). \qquad (4)$$

In that sense, MIG effectively captures the pairwise interactions between latent variables. This metric still ignores interplays between more than two variables. Figure 2b reveals that some of the redundant information $\mathcal{R}(y_k; z_\ell, \mathbf{z}_{\setminus \ell})$ is positively evaluated in MIG, which should have been considered as a signal of entanglement. Note that there are several extensions to MIG (Do & Tran, 2020; Li et al., 2020; Zaidi et al., 2020); see Appendix B for detailed discussions on them.

---

[2]Most studies on PID only deal with discrete systems, while deep representations often include continuous variables. There have been attempts to define and analyze PID for continuous systems (Barrett, 2015; Schick-Poland et al., 2021; Pakman et al., 2021). Note that the domain of variables, which our framework depends on, is not limited with the PID framework.

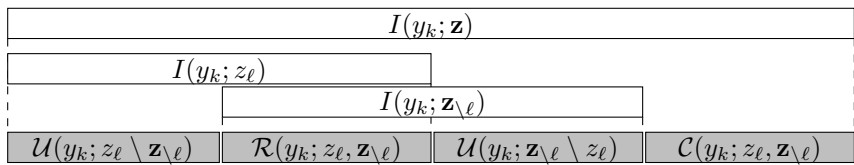

(a) PID for systems with $y_k$, $z_\ell$, and $\mathbf{z}_{\backslash \ell}$

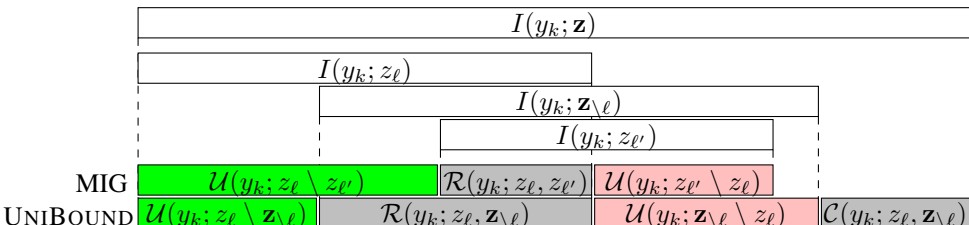

(b) Side-by-side comparison of positive and negative terms in UNIBOUND and MIG

Figure 2: Information diagrams depicted by bands (the style borrowed from Figure 8.1 of MacKay (2003)). White boxes represent mutual information, which we can compute. (**a**) The bands depict the decomposition used in the PID-based disentanglement evaluation. (**b**) This diagram superposes the decomposition for systems with $(y_k, z_\ell, \mathbf{z}_{\backslash \ell})$ and $(y_k, z_\ell, z_{\ell'})$ where $z_{\ell'}$ is the latent variable chosen by MIG evaluation. The green boxes are positively evaluated in MIG (the top colored line) and UNIBOUND (the bottom colored line), while the red boxes are negatively evaluated in them. Observe that MIG positively evaluates a part of the redundancy, namely $\mathcal{R}(y_k; z_\ell, \mathbf{z}_{\backslash \ell}) - \mathcal{R}(y_k; z_\ell, z_{\ell'})$, as it does not take into account the interactions among strict supersets of $\{z_\ell, z_{\ell'}\}$.

## 3.3 UNIBOUND: NOVEL DISENTANGLEMENT METRIC

We can lower bound the unique information in any possible PID definitions by computable components, as we did in Eq.4. To bound $\mathcal{U}(y_k; z_\ell \backslash \mathbf{z}_{\backslash \ell})$ instead of $\mathcal{U}(y_k; z_\ell \backslash z_{\ell'})$, we replace $z_{\ell'}$ with $\mathbf{z}_{\backslash \ell}$, obtaining

$$\mathcal{U}(y_k; z_\ell \backslash \mathbf{z}_{\backslash \ell}) \geq \left[\mathcal{U}(y_k; z_\ell \backslash \mathbf{z}_{\backslash \ell}) - \mathcal{U}(y_k; \mathbf{z}_{\backslash \ell} \backslash z_\ell)\right]_+ = \left[I(y_k; z_\ell) - I(y_k; \mathbf{z}_{\backslash \ell})\right]_+ \quad (5)$$

where we use $[\cdot]_+ = \max\{0, \cdot\}$ as the difference in mutual information is not guaranteed to be non-negative. The decomposed terms evaluated by the bound is illustrated in the lower part of Figure 2b. It effectively excludes, from the positive term, the effect of interaction between $z_\ell$ and any other latent variables. In a similar way to MIG, we summarize this bound over all the generative factors to obtain the metric we call UNIBOUND.

$$\text{UNIBOUND} := \frac{1}{K} \sum_{k=1}^{K} \frac{1}{H(y_k)} \max_\ell \left[I(y_k; z_\ell) - I(y_k; \mathbf{z}_{\backslash \ell})\right]_+. \quad (6)$$

Dividing each summand by the entropy $H(y_k)$ has the same role as in MIG; it makes the evaluation fair between factors and eases the comparison as the metric is normalized when $y_k$ is discrete.

## 3.4 OTHER BOUNDS FOR PARTIAL INFORMATION TERMS

The UNIBOUND metric is a handy quantity to compare representations by a single scalar, while PID itself may provide more ideas on how a given representation entangles or disentangles the factors. To fully leverage the potential, we derive bounds for all the terms of interest, including redundancy and synergy terms, from both lower and upper sides.

Let $II(y_k; z_\ell; \mathbf{z}_{\backslash \ell}) = I(y_k; z_\ell) + I(y_k; \mathbf{z}_{\backslash \ell}) - I(y_k; \mathbf{z})$ be the interaction information of a triple $(y_k, z_\ell, \mathbf{z}_{\backslash \ell})$. Using nonnegativity of PID terms, we can derive the following bounds from Eq.1-2.

$$\left[I(y_k; z_\ell) - I(y_k; \mathbf{z}_{\backslash \ell})\right]_+ \leq \mathcal{U}(y_k; z_\ell \backslash \mathbf{z}_{\backslash \ell}) \leq I(y_k; z_\ell) - \left[II(y_k; z_\ell; \mathbf{z}_{\backslash \ell})\right]_+,$$

$$\left[II(y_k; z_\ell; \mathbf{z}_{\backslash \ell})\right]_+ \leq \mathcal{R}(y_k; z_\ell, \mathbf{z}_{\backslash \ell}) \leq \min\{I(y_k; z_\ell), I(y_k; \mathbf{z}_{\backslash \ell})\}, \quad (7)$$

$$\left[-II(y_k; z_\ell; \mathbf{z}_{\backslash \ell})\right]_+ \leq \mathcal{C}(y_k; z_\ell, \mathbf{z}_{\backslash \ell}) \leq \min\{I(y_k; z_\ell), I(y_k; \mathbf{z}_{\backslash \ell})\} - II(y_k; z_\ell; \mathbf{z}_{\backslash \ell}).$$

Note that all six bounds are computed by arithmetics on $I(y_k; z_\ell)$, $I(y_k; \mathbf{z}_{\setminus \ell})$, and $I(y_k; \mathbf{z})$. We can summarize each lower bound in a similar way as we did in Eq.6 and summarize the corresponding upper bound using the same $\ell$ for each $k$ as the lower bound. While these bounds only determine the terms as intervals, they provide us enough insight into the type of entanglement dominant in the representation (redundancy or synergy).

### 3.5 Estimating Bounds by Exact Log-Marginal Densities

When the dataset is fully annotated with discrete generative factors and the inference distribution $p(\mathbf{z}|x)$ and its marginals $p(z_\ell|x), p(\mathbf{z}_{\setminus \ell}|x)$ are all tractable (e.g., mean field variational models), we can compute the bounds in a similar way as is done by Chen et al. (2018) for MIG. Let $\mathbf{z}_S$ be either of $z_\ell, \mathbf{z}_{\setminus \ell}$ or $\mathbf{z}$. We denote by $\mathcal{D}(y_k)$ the subset of the dataset $\mathcal{D}$ with a specific value of $y_k$. The mutual information $I(y_k; \mathbf{z}_S) = -H(\mathbf{z}_S|y_k) + H(\mathbf{z}_S)$ can then be computed by the following formula.

$$I(y_k; \mathbf{z}_S) = \mathbb{E}_{p(y_k, \mathbf{z}_S)} \left[ \log \frac{1}{|\mathcal{D}(y_k)|} \sum_{x \in \mathcal{D}(y_k)} p(\mathbf{z}_S|x) \right] - \mathbb{E}_{p(y_k, \mathbf{z}_S)} \left[ \log \frac{1}{|\mathcal{D}|} \sum_{x \in \mathcal{D}} p(\mathbf{z}_S|x) \right], \quad (8)$$

Assuming that each generative factor $y_k$ is discrete and uniform, we employ stratified sampling over $p(y_k, \mathbf{z}_S)$. We approximate the expectation over $p(\mathbf{z}_S|y_k)$ by sampling $x$ from the subset $\mathcal{D}(y_k)$ and then sampling $\mathbf{z}_S$ from $p(\mathbf{z}|x)$ to avoid quadratic computational cost. Following Chen et al. (2018), we used the sample size of 10000 in experiments. We use log-sum-exp function to compute $\log \sum p(\mathbf{z}_S|x) = \log \sum \exp(\log p(\mathbf{z}_S|x))$ for numerical stability. The PID bounds and the UniBound metric are computed by combining these estimations.

When the inference distribution $p(\mathbf{z}|x)$ is factorized, its log marginal $\log p(\mathbf{z}_S|x)$ is computed by just adding up the log marginal of each variable as $\sum_{\ell \in S} \log p(z_\ell|x)$. Otherwise, we need to explicitly derive the marginal distribution and compute the log density. For example, when $p(\mathbf{z}|x)$ is a Gaussian distribution with mean $\boldsymbol{\mu}$ and non-diagonal covariance $\boldsymbol{\Sigma}$, $p(\mathbf{z}_S|x)$ is a Gaussian distribution with mean $(\mu_i)_{i \in S}$ and covariance $(\Sigma_{ij})_{i,j \in S}$. Such a case arises for the attacked model we will describe in the next section.

Let $M$ be the sample size for the expectations, and $N = |\mathcal{D}|$ be the size of the dataset. Then, the computation of Eq.8 requires $O(MN)$ evaluations of the conditional density $p(\mathbf{z}_S|x)$.

## 4 Entanglement Attacks

To confirm that the proposed framework effectively captures interactions among multiple latent variables, we apply it to adversarial representations that entangle any generative factors. Instead of making a single artificial model, we modify a given model that disentangles factors well into a noised version that entangles the original variables. We call this process an *entanglement attack*.

Let $\mathbf{z} \in \mathbb{R}^L$ be the representation defined by a given model. Our goal is to design a transform from $\mathbf{z}$ to an attacked representation $\tilde{\mathbf{z}}$ so that disentanglement metrics fail to capture entanglement unless they correctly handle multi-variable interactions.

As we mentioned in Section 3, metrics that do not take into account the interaction among multiple latent variables may underestimate redundant information. To crystallize such a situation, we first design an entanglement attack to inject redundant information into multiple variables. For completeness, we also design a similar attack to inject synergistic information as well.

### 4.1 Redundancy Attack

Let $\mathbf{U}$ be an $L \times L$ orthonormal matrix and $\boldsymbol{\epsilon} \in \mathbb{R}^L$ be a random vector following the standard normal distribution $\mathcal{N}(\mathbf{0}, \mathbf{I})$. Using a hyperparameter $\alpha \geq 0$, we define the *redundancy attack* by

$$\tilde{\mathbf{z}}^{\text{red}} = \begin{pmatrix} \mathbf{z} \\ \alpha \mathbf{U} \mathbf{z} + \boldsymbol{\epsilon} \end{pmatrix}. \quad (9)$$

The coefficient $\alpha$ adjusts the degree of entanglement. When $\alpha = 0$, the new representation just appends noise elements to a given representation, which does not affect the disentanglement. Increasing $\alpha$ makes the additional dimensions less noisy, resulting in $\tilde{\mathbf{z}}^{\text{red}}$ redundantly encoding the information of factors conveyed by the original representation. The mixing matrix $\mathbf{U}$ chooses how the information of individual variables in $\mathbf{z}$ is distributed to the additional dimensions. If we choose

$\mathbf{U} = \mathbf{I}$, the dimensions are not mixed; thus considering one-to-one interaction between pairs of variables is enough to capture the redundancy. To mix the dimensions, we can use $\mathbf{U} = \mathbf{I} - \frac{2}{L}\mathbf{1}\mathbf{1}^\top$ instead, which is an orthonormal matrix that mixes each variable with the others.

To evaluate mutual information terms for MIG and UNIBOUND metrics after the attack, we need an explicit formula for the inference distribution $p(\tilde{\mathbf{z}}^{\mathrm{red}}|x)$. When the original model has a Gaussian inference distribution $p(\mathbf{z}|x) = \mathcal{N}(\mathbf{z}; \boldsymbol{\mu}(x), \boldsymbol{\Sigma}(x))$, the attacked model is a summation of linear transformation of $\mathbf{z}$ and a standard normal noise, which results in a Gaussian distribution

$$p(\tilde{\mathbf{z}}^{\mathrm{red}}|x) = \mathcal{N}\left(\tilde{\mathbf{z}}^{\mathrm{red}}; \begin{pmatrix} \boldsymbol{\mu}(x) \\ \alpha\mathbf{U}\boldsymbol{\mu}(x) \end{pmatrix}, \begin{pmatrix} \boldsymbol{\Sigma}(x) & \alpha\boldsymbol{\Sigma}(x)\mathbf{U}^\top \\ \alpha\mathbf{U}\boldsymbol{\Sigma}(x) & \mathbf{I} + \alpha^2\mathbf{U}\boldsymbol{\Sigma}(x)\mathbf{U}^\top \end{pmatrix}\right).$$

## 4.2 SYNERGY ATTACK

For completeness, we also define an attack that entangles representation by increasing synergistic information. Using the same setting with an $L \times L$ orthonormal matrix $\mathbf{U}$ and a noise vector $\boldsymbol{\epsilon} \sim \mathcal{N}(\mathbf{0}, \mathbf{I})$, we construct the *synergy attack* by

$$\tilde{\mathbf{z}}^{\mathrm{syn}} = \begin{pmatrix} \alpha\mathbf{U}\boldsymbol{\epsilon} + \mathbf{z} \\ \boldsymbol{\epsilon} \end{pmatrix}. \tag{10}$$

Here again, the coefficient $\alpha$ adjusts the degree of entanglement. When $\alpha = 0$, the attacked version just extends the original representation with independent noise elements. By increasing $\alpha$, the noise are mixed with the parts conveying the information of factors. The attacked vector $\tilde{\mathbf{z}}^{\mathrm{syn}}$ fully conveys the information of factors in $\mathbf{z}$ regardless of $\alpha$, as we can recover the original representation by $\mathbf{z} = \tilde{\mathbf{z}}^{\mathrm{syn}}_{1:L} - \alpha\mathbf{U}\tilde{\mathbf{z}}^{\mathrm{syn}}_{L+1:2L}$. Note that most existing metrics correctly react to this attack since the information of individual variables is destroyed by the noise. The upper bound of the unique information is one of the quantities that positively evaluate synergistic information and is expected to ignore the attack.

## 5 EVALUATION

We evaluated the metrics in a toy model in Section 5.1 and in VAEs trained on datasets with the true generative factors in Section 5.2. We also performed detailed analyses of VAEs by plotting the PID terms of each factor, which is deferred to Appendix E due to the limited space.

## 5.1 EXACT ANALYSIS WITH TOY MODEL

We consider a toy model with attacks defined in Section 4 as a sanity check. Suppose that data are generated by factors $\mathbf{y} \sim \mathcal{N}(\mathbf{0}, \mathbf{I})$, and we have a latent representation that disentangles them up to noise as $\mathbf{z}|\mathbf{y} \sim \mathcal{N}(\mathbf{y}, \sigma^2\mathbf{I})$, where $\sigma > 0$. With this simple setting, we can analytically compute MIG and UNIBOUND. For example, when we set $\sigma = 0.1$ and $K = 5$, we can derive the scores after the redundancy attack as $\mathrm{MIG} = \frac{1}{c}\log\left(101 \times \frac{1+0.65\alpha^2}{1+1.01\alpha^2}\right)$ and $\mathrm{UNIBOUND} = \frac{1}{c}\log\left(101 \times \frac{1+0.01\alpha^2}{1+1.01\alpha^2}\right)$, where $c = \log(2\pi e)$. As a function of $\alpha$, UNIBOUND decreases faster than MIG. The difference comes from the information distributed among the added dimensions of the attacked vector; while UNIBOUND counts all the entangled information, MIG only deals with one of the added dimensions. Indeed, the amount of the untracked entanglement remains in MIG after taking the limit of $\alpha \to \infty$ as $\mathrm{MIG} \to \frac{1}{c}\log 65$, while $\mathrm{UNIBOUND} \to 0$.

We can also compute the scores after the synergy attack as $\mathrm{MIG} = \mathrm{UNIBOUND} = \frac{1}{c}\log\left(1 + \frac{1}{\alpha^2+0.01}\right)$, where both metrics correctly capture the injected synergy. Refer to Appendix C for the derivations of the exact scores for arbitrary parameter choice.

## 5.2 EMPIRICAL ANALYSIS WITH ANNOTATED DATASETS

We used the DSPRITES and 3DSHAPES datasets for our analysis. The DSPRITES dataset consists of $737,280$ binary images generated from five generative factors (shape, size, rotation, and x/y coordinates). The 3DSHAPES dataset consists of $480,000$ color images generated from six generative factors (floor/wall/object hues, scale, shape, and orientation). All the images have $64 \times 64$ pixels. We used all the factors for $\mathbf{y}$, encoded as a set of discrete (categorical) random variables.

We used variants of VAEs as methods for disentangled representation learning: $\beta$-VAE (Higgins et al., 2017), FactorVAE (Kim & Mnih, 2018), $\beta$-TCVAE (Chen et al., 2018), and JointVAE (Dupont, 2018). We trained all the models with six latent variables, one of which in JointVAE is

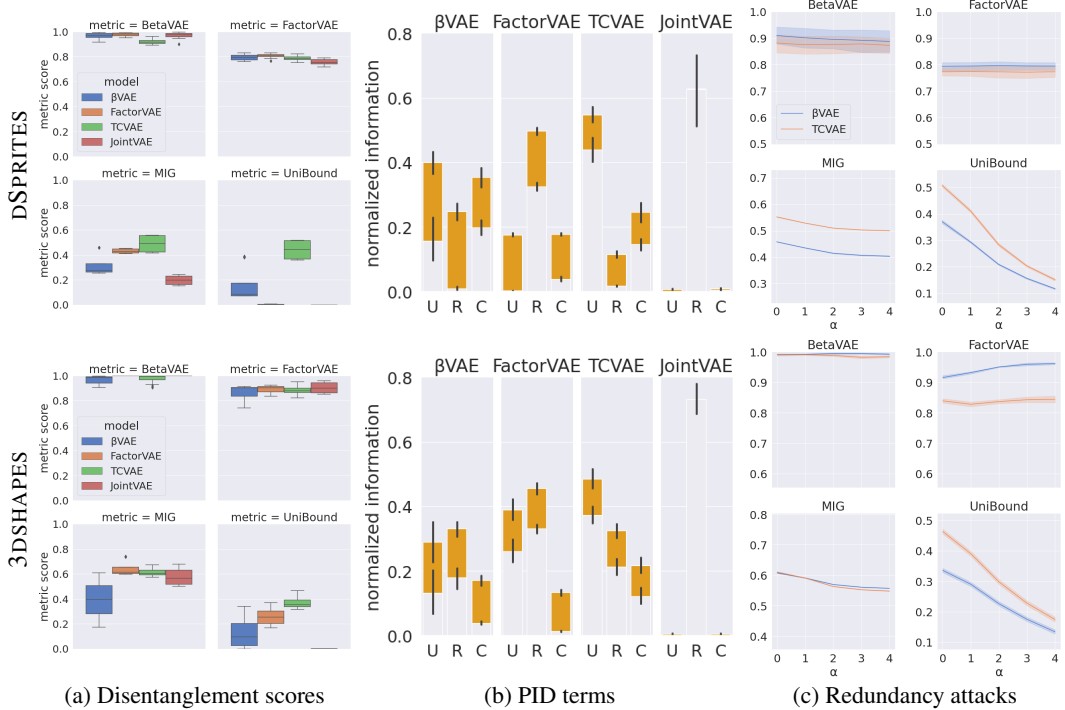

Figure 3: Experimental results for VAEs trained with DSPRITES (top row) and 3DSHAPES (bottom row). **(a)** Disentanglement scores. **(b)** Estimated PID terms. Three orange bars in each plot represent the possible values of unique (U), redundant (R), and complementary (C) information, respectively. The top and bottom of each orange area correspond to the upper and lower bounds of the term, computed with Eq.7. **(c)** Disentanglement scores of $\beta$-VAE and $\beta$-TCVAE after redundancy attack with varying strength. See Appendix H for a larger version of the plots.

a three-way categorical variable for DSPRITES and a four-way categorical variable for 3DSHAPES. We trained each model eight times with different random seeds and chose the best half of them for each metric to avoid cluttered results due to training instability. We optimized network weights with Adam (Kingma & Ba, 2015). We used the standard convolutional networks used in the literature for the encoder and the decoder. See Appendix D for the details of architectures and hyperparameters.

For disentanglement metrics, we compare BetaVAE metric (Higgins et al., 2017), FactorVAE metric (Kim & Mnih, 2018), MIG (Chen et al., 2018), and the proposed UNIBOUND metric.

We first compared the models with each metric as shown in Figure 3a. The trend is basically similar between UNIBOUND and the other metrics, while they disagree in some cases. For example, Factor-VAE achieves a higher MIG score for DSPRITES than $\beta$-VAE, while its UNIBOUND score is low. As we saw in Figure 2b, such a case occurs when a part of the redundancy $\mathcal{R}(y_k; z_\ell, \mathbf{z}_{\setminus \ell}) - \mathcal{R}(y_k; z_\ell, z_{\ell'})$ is large. This observation indicates that FactorVAE effectively forces each variable to encode information of distinct factors (i.e., one-vs-one redundancy is small), while it fails to avoid entangling the information over multiple variables (i.e., one-vs-all redundancy is large).

We can confirm that the redundancy is indeed large in FactorVAE by computing the PID bounds. In Figure 3b, we plot the aggregated bounds of $\mathcal{U}(y_k; z_\ell \setminus \mathbf{z}_{\setminus \ell})$, $\mathcal{R}(y_k; z_\ell, \mathbf{z}_{\setminus \ell})$, and $\mathcal{C}(y_k; z_\ell, \mathbf{z}_{\setminus \ell})$. The plot reveals that FactorVAE tends to encode the factors redundantly into multiple latent variables.

To understand the tasks and models more deeply, we evaluated the PID bounds for each factor in Figure 4. We can see that the factors are captured by the models in various ways. For example, $\beta$-TCVAE succeeds to disentangle the position and scale factors in DSPRITES, while it encodes orientation synergistically. It may reflect the inherent difficulty of disentangling this factor in DSPRITES, as the image of each shape corresponding to $0°$ orientation is chosen arbitrarily. These observations help us to choose what kind of inductive biases to introduce into the models.

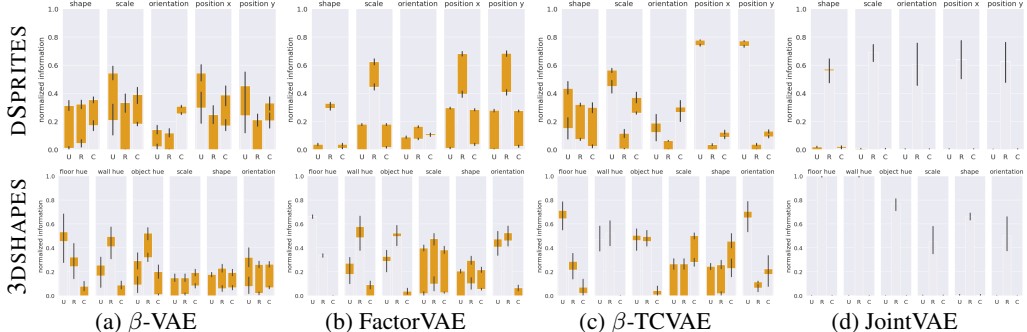

Figure 4: Estimated PID terms of each factor in DSPRITES and 3DSHAPES. As in Figure 3b, three orange bars in each plot show the range between lower and upper bound estimations of unique information (U), redundant information (R), and complementary information (C). See Figure 6 in Appendix for a larger version and Table 5 for qualitative interpretations.

FactorVAE, on the other hand, tends to encode factors redundantly. This indicates FactorVAE succeeds to make individual variables encode each factor, while it fails to prevent other variables from encoding the same information. JointVAE also encodes factors redundantly. While it fails to disentangle all the factors, this is the only model that encodes the shape and orientation to single latent variables. This can be viewed as the effect of introducing a discrete variable into the representation.

We can understand large redundancy as the models failing to make variables independent enough. The lower bound of redundancy in Eq.7 is related to independence as

$$[II(y_k; z_\ell; \mathbf{z}_{\setminus \ell})]_+ = [I(z_k; \mathbf{z}_{\setminus \ell}) - I(z_\ell; \mathbf{z}_{\setminus \ell}|y_k)]_+ \leq I(z_k; \mathbf{z}_{\setminus \ell}). \tag{11}$$

Therefore, a high redundancy lower bound indicates large mutual information $I(z_k; \mathbf{z}_{\setminus \ell})$; i.e., the latent variables are highly dependent. We conjecture that FactorVAE, which approximates the total correlation $D_{\mathrm{KL}}(p(\mathbf{z})\| \prod_\ell p(z_\ell))$ by a critic, fails to make the critic capture the dependency in $\mathbf{z}$ enough in our experiments. Since the MIG score is relatively high, the critic succeeds to capture pairwise dependency, while it fails to capture higher dimensional dependency. The high redundancy in JointVAE can also be explained by the lack of independence; see Appendix F for details.

As in Figure 3a, some metrics have large deviations from the median. This is caused by the randomness in training rather than in evaluation; see Appendix I for detailed analyses.

As we did for the toy model, we performed entanglement attacks in Figure 3c to assess its effect on each metric in learned representations. We selected the best training trials of $\beta$-VAE and $\beta$-TCVAE. Here, we only plot the results with the redundancy attack, as each metric already behaves well against the synergy attack. The plot reveals that BetaVAE and FactorVAE metrics do not detect the redundancy injected by the attack. MIG slightly decreases with the attack, which is not significant against the score variation between learning methods as we observed in Figure 3a. UNIBOUND strongly reacts against the attack, indicating that it effectively detects the injected redundancy.

## 6 CONCLUSION

We established a framework of disentanglement analysis using Partial Information Decomposition. We formulated a new disentanglement metric, UNIBOUND, using the unique information bounds, and confirmed with entanglement attacks that the metric correctly responses to entanglement caused by multi-variable interactions which are not captured by other metrics. UNIBOUND sometimes disagrees with other metrics on VAEs trained with controlled datasets, which indicates that multi-variable interactions arise not only in artificial settings but in learned representations. We found that VAEs trained with different methods induce representations with a variety of ratios between PID terms, even if their disentanglement scores are close. It is a major future work to develop learning methods of disentangled representations on the basis of these observations.

ACKNOWLEDGMENTS

We thank members of Issei Sato Laboratory and researchers in Preferred Networks for fruitful discussions, and thank the reviewers for helpful comments to improve the work.

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

## A    ADDITIONAL DIAGRAMS

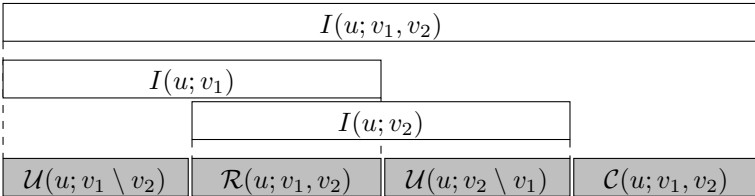

Figure 5: Alternative diagram of Figure 1 in the band style following Figure 8.1 of MacKay (2003).

## B    RELATIONSHIPS TO OTHER INFORMATION-THEORETIC METRICS

There have been several metrics proposed in the literature for information-theoretic measurement of disentanglement.

For example, Do & Tran (2020) proposed four metrics, namely WSEPIN, WINDIN, RMIG, and JEMMIG. Among them, WSEPIN is computed based on the information gap $I(x; z_\ell | \mathbf{z}_{\backslash \ell}) = I(x; \mathbf{z}) - I(x; \mathbf{z}_\ell)$. In the PID perspective, this quantity upper bounds the unique information of $x$ held by $z_\ell$, i.e., $I(x; z_\ell | \mathbf{z}_{\backslash \ell}) \geq \mathcal{U}(x; z_\ell \setminus \mathbf{z}_{\backslash \ell})$. It is similar to the upper bound of unique information that we derived in Eq.7, as

$$I(y_k; z_\ell) - [II(y_k; z_\ell; \mathbf{z}_{\backslash \ell})]_+ \leq I(y_k; z_\ell) - II(y_k; z_\ell; \mathbf{z}_{\backslash \ell}) = I(y_k; \mathbf{z}) - I(y_k; \mathbf{z}_{\backslash \ell}). \tag{12}$$

In that sense, WSEPIN is an upper bound of unique information, where $y_k$ is replaced with $x$ and the nonnegativity of redundant information is ignored. As WSEPIN does not handle generative factors separately, it is useful when the generative factors are unknown, while our approach provides more detailed information of (dis)entanglement when the generative factors are available.

RMIG is an extension to MIG, where the conditioning between $y_k$ and $x$ is inverted so that one can compute the quantity with uncontrolled dataset. This approach may be extended for our framework as well, in a similar way to applying the inversion to MIG. We did not use this extension as we only deal with controlled datasets in this paper to contrast our approach with a wider variety of metrics.

JEMMIG is also an extention to MIG, which provides a measurement of how each variable captures only one generative factor. This aspect is also studied by Eastwood & Williams (2018) and Li et al. (2020). As we used a set of simple latent variables each of which is either a one-dimensional real variable or a categorical variable with a small number of arms (three or four depending on the dataset), we expect that each variable does not capture much information of multiple generative factors. Indeed, we did not observe any latent variable selected for more than two generative factors, and when a latent variable is selected for two generative factors, the overall disentanglement score is low so that the effect of duplicated selection is ignorable. We still consider it an interesting future work to extend our approach for analyzing entanglement of multiple generative factors into a single latent variable.

## C    DERIVATIONS FOR THE EXACT ANALYSIS WITH THE TOY MODEL

In this section, we provide a sketch of analytically computing the metrics for the toy model we used in Section 5.1. We use the following formulae.

- Let $\mathbf{u}$ be a Gaussian vector with covariance $\boldsymbol{\Sigma}$. Then, its entropy is given by $H(\mathbf{u}) = \frac{1}{2} \log(2\pi e |\boldsymbol{\Sigma}|)$, where $|\cdot|$ is the matrix determinant. When we remove the $\ell$-th element from $\mathbf{u}$, the remaining vector $\mathbf{u}_{\backslash \ell}$ follows a Gaussian distribution whose covariance matrix $\boldsymbol{\Sigma}_{\backslash \ell}$ is obtained by removing the $\ell$-th row and column from $\boldsymbol{\Sigma}$. The determinant of $\boldsymbol{\Sigma}_{\backslash \ell}$ is the cofactor of $\boldsymbol{\Sigma}$ at the $\ell$-th diagonal element, i.e., $|\boldsymbol{\Sigma}_{\backslash \ell}| = |\boldsymbol{\Sigma}| (\boldsymbol{\Sigma}^{-1})_{\ell\ell}$. With this formula, we can derive the entropy of $\mathbf{u}_{\backslash \ell}$ as $H(\mathbf{u}_{\backslash \ell}) = H(\mathbf{u}) + \frac{1}{2} \log(\boldsymbol{\Sigma}^{-1})_{\ell\ell}$.

- Suppose that an invertible matrix $\boldsymbol{\Sigma}$ is written by blocks as $\boldsymbol{\Sigma} = \begin{pmatrix} \mathbf{A} & \mathbf{B} \\ \mathbf{C} & \mathbf{D} \end{pmatrix}$. If $\mathbf{A}$ and its Schur complement $\mathbf{S} := \mathbf{D} - \mathbf{C} \mathbf{A}^{-1} \mathbf{B}$ are invertible, the determinant and inverse of $\boldsymbol{\Sigma}$ are

Table 1: Exact values of metrics for toy model. We use $\mathbf{U} = \mathbf{I} - \frac{2}{K}\mathbf{1}\mathbf{1}^\top$ for the attacks. We do not normalize the metrics by $H(y_k)$ as $\mathbf{y}$ is isotropic and continuous; normalization by it just scales the results by a constant scalar. For the attacked models, we show the limit of $\alpha \to \infty$ to understand how well each metric reacts to completely entangled representations.

| Metric | $\mathbf{z}$ | $\tilde{\mathbf{z}}^{\text{red}}$ | $\tilde{\mathbf{z}}^{\text{syn}}$ |
|---|---|---|---|
| MIG | $\frac{1}{2}\log\left(1 + \frac{1}{\sigma^2}\right)$ | $\frac{1}{2}\log\frac{(1+\sigma^2)(1+\alpha^2(1+\sigma^2-(1-\frac{2}{K})^2))}{\sigma^2(1+\alpha^2(1+\sigma^2))}$ $\to \frac{1}{2}\log\left(1 + \frac{1-(1-\frac{2}{K})^2}{\sigma^2}\right)$ | $\frac{1}{2}\log\left(1 + \frac{1}{\alpha^2+\sigma^2}\right) \to 0$ |
| $\mathcal{U}$ lower bound (UNIBOUND) | $\frac{1}{2}\log\left(1 + \frac{1}{\sigma^2}\right)$ | $\frac{1}{2}\log\frac{(1+\sigma^2)(1+\alpha^2\sigma^2)}{\sigma^2(1+\alpha^2(1+\sigma^2))} \to 0$ | $\frac{1}{2}\log\left(1 + \frac{1}{\alpha^2+\sigma^2}\right) \to 0$ |
| $\mathcal{U}$ upper bound | $\frac{1}{2}\log\left(1 + \frac{1}{\sigma^2}\right)$ | $\frac{1}{2}\log\frac{(1+\sigma^2)(1+\alpha^2\sigma^2)}{\sigma^2(1+\alpha^2(1+\sigma^2))} \to 0$ | $\frac{1}{2}\log\left(1 + \frac{1}{\alpha^2+\sigma^2}\right) \to 0$ |

written as

$$|\mathbf{\Sigma}| = |\mathbf{A}| \cdot |\mathbf{S}|, \quad \mathbf{\Sigma}^{-1} = \begin{pmatrix} \mathbf{A}^{-1} + \mathbf{A}^{-1}\mathbf{B}\mathbf{S}^{-1}\mathbf{C}\mathbf{A}^{-1} & -\mathbf{A}^{-1}\mathbf{B}\mathbf{S}^{-1} \\ -\mathbf{S}^{-1}\mathbf{C}\mathbf{A}^{-1} & \mathbf{S}^{-1} \end{pmatrix}. \quad (13)$$

Similarly, if $\mathbf{D}$ and its Schur complement $\mathbf{T} := \mathbf{A} - \mathbf{B}\mathbf{D}^{-1}\mathbf{C}$ are invertible, then

$$|\mathbf{\Sigma}| = |\mathbf{D}| \cdot |\mathbf{T}|, \quad \mathbf{\Sigma}^{-1} = \begin{pmatrix} \mathbf{T}^{-1} & -\mathbf{T}^{-1}\mathbf{B}\mathbf{D}^{-1} \\ -\mathbf{D}^{-1}\mathbf{C}\mathbf{T}^{-1} & \mathbf{D}^{-1} + \mathbf{D}^{-1}\mathbf{C}\mathbf{T}^{-1}\mathbf{B}\mathbf{D}^{-1} \end{pmatrix}. \quad (14)$$

Provided that $\mathbf{z}$ is a Gaussian vector with covariance $\mathbf{\Sigma}$, we can derive the entropies of the attacked vectors. Recall that the redundancy and synergy attacks are defined as follows.

$$\tilde{\mathbf{z}}^{\text{red}} = \begin{pmatrix} \mathbf{I} & \mathbf{0} \\ \alpha\mathbf{U} & \mathbf{I} \end{pmatrix}\begin{pmatrix} \mathbf{z} \\ \boldsymbol{\epsilon} \end{pmatrix}, \quad \tilde{\mathbf{z}}^{\text{syn}} = \begin{pmatrix} \mathbf{I} & \alpha\mathbf{U} \\ \mathbf{0} & \mathbf{I} \end{pmatrix}\begin{pmatrix} \mathbf{z} \\ \boldsymbol{\epsilon} \end{pmatrix}$$

where $\mathbf{U}$ is an orthonormal matrix, $\alpha > 0$, and $\boldsymbol{\epsilon} \sim \mathcal{N}(\mathbf{0}, I)$. Here, $\mathbf{0}$ is the zero matrix. These again follow Gaussian distributions, whose covariance matrices are computed as follows.

$$\text{Cov}(\tilde{\mathbf{z}}^{\text{red}}) = \begin{pmatrix} \mathbf{I} & \mathbf{0} \\ \alpha\mathbf{U} & \mathbf{I} \end{pmatrix}\begin{pmatrix} \mathbf{\Sigma} & \mathbf{0} \\ \mathbf{0} & \mathbf{I} \end{pmatrix}\begin{pmatrix} \mathbf{I} & \alpha\mathbf{U}^\top \\ \mathbf{0} & \mathbf{I} \end{pmatrix} = \begin{pmatrix} \mathbf{\Sigma} & \alpha\mathbf{\Sigma}\mathbf{U}^\top \\ \alpha\mathbf{U}\mathbf{\Sigma} & \mathbf{I} + \alpha^2\mathbf{U}\mathbf{\Sigma}\mathbf{U}^\top \end{pmatrix},$$

$$\text{Cov}(\tilde{\mathbf{z}}^{\text{syn}}) = \begin{pmatrix} \mathbf{I} & \alpha\mathbf{U} \\ \mathbf{0} & \mathbf{I} \end{pmatrix}\begin{pmatrix} \mathbf{\Sigma} & \mathbf{0} \\ \mathbf{0} & \mathbf{I} \end{pmatrix}\begin{pmatrix} \mathbf{I} & \mathbf{0} \\ \alpha\mathbf{U}^\top & \mathbf{I} \end{pmatrix} = \begin{pmatrix} \alpha^2\mathbf{I} + \mathbf{\Sigma} & \alpha\mathbf{U} \\ \alpha\mathbf{U}^\top & \mathbf{I} \end{pmatrix}.$$

Using Eq.13 and Eq.14, we obtain their determinants and inverses.

$$\left|\text{Cov}(\tilde{\mathbf{z}}^{\text{red}})\right| = |\mathbf{\Sigma}|, \qquad \text{Cov}(\tilde{\mathbf{z}}^{\text{red}})^{-1} = \begin{pmatrix} \alpha^2\mathbf{I} + \mathbf{\Sigma}^{-1} & -\alpha\mathbf{U}^\top \\ -\alpha\mathbf{U} & \mathbf{I} \end{pmatrix},$$

$$\left|\text{Cov}(\tilde{\mathbf{z}}^{\text{syn}})\right| = |\mathbf{\Sigma}|, \qquad \text{Cov}(\tilde{\mathbf{z}}^{\text{syn}})^{-1} = \begin{pmatrix} \mathbf{\Sigma}^{-1} & -\alpha\mathbf{\Sigma}^{-1}\mathbf{U} \\ -\alpha\mathbf{U}^\top\mathbf{\Sigma}^{-1} & \mathbf{I} + \alpha^2\mathbf{U}^\top\mathbf{\Sigma}^{-1}\mathbf{U} \end{pmatrix}.$$

Therefore, we obtain the following entropies for each $\ell \in \{1, \ldots, L\}$.

$$H(\tilde{z}_\ell^{\text{red}}) = \frac{1}{2}\log(2\pi e\mathbf{\Sigma}_{\ell\ell}), \qquad\qquad H(\tilde{\mathbf{z}}_{\backslash\ell}^{\text{red}}) = \frac{1}{2}\log(2\pi e|\mathbf{\Sigma}|(\alpha^2 + (\mathbf{\Sigma}^{-1})_{\ell\ell})),$$

$$H(\tilde{z}_{L+\ell}^{\text{red}}) = \frac{1}{2}\log(2\pi e(1 + \alpha^2\mathbf{u}_\ell^\top\mathbf{\Sigma}\mathbf{u}_\ell)), \quad H(\tilde{\mathbf{z}}_{\backslash L+\ell}^{\text{red}}) = \frac{1}{2}\log(2\pi e|\mathbf{\Sigma}|),$$

$$H(\tilde{z}_\ell^{\text{syn}}) = \frac{1}{2}\log(2\pi e(\alpha^2 + \mathbf{\Sigma}_{\ell\ell})), \qquad\qquad H(\tilde{\mathbf{z}}_{\backslash\ell}^{\text{syn}}) = \frac{1}{2}\log(2\pi e|\mathbf{\Sigma}|(\mathbf{\Sigma}^{-1})_{\ell\ell}),$$

$$H(\tilde{z}_{L+\ell}^{\text{syn}}) = \frac{1}{2}\log(2\pi e), \qquad\qquad H(\tilde{\mathbf{z}}_{\backslash L+\ell}^{\text{syn}}) = \frac{1}{2}\log(2\pi e|\mathbf{\Sigma}|(1 + \alpha^2\mathbf{v}_\ell^\top\mathbf{\Sigma}^{-1}\mathbf{v}_\ell)),$$

$$H(\tilde{\mathbf{z}}^{\text{red}}) = H(\tilde{\mathbf{z}}^{\text{syn}}) = \frac{1}{2}\log(2\pi e|\mathbf{\Sigma}|).$$

$$(15)$$

Here, $\mathbf{u}_\ell$ and $\mathbf{v}_\ell$ are the $\ell$-th columns of $\mathbf{U}^\top$ and $\mathbf{U}$, respectively. When we use $\mathbf{U} = \mathbf{I} - \frac{2}{K}\mathbf{1}\mathbf{1}^\top$, these are written as $\mathbf{u}_\ell = \mathbf{v}_\ell = \mathbf{e}_\ell - \frac{2}{K}\mathbf{1}$, where $(\mathbf{e}_1, \ldots, \mathbf{e}_L)$ is the standard basis of $\mathbb{R}^L$.

Table 2: Encoder and decoder architectures used in the DSPRITES and 3DSHAPES experiments. Data flow top to bottom. Conv4x4s2p1 represents spatial convolution layer with kernel size 4x4, stride 2x2, and 1 pixel padding at each side of the image. ConvT represents the transposed convolution layer. FC stands for a fully-connected layer. The last fully-connected layer of the encoder outputs the parameters of the variational posterior distribution, the number of which depends on the model definition as follows. We used six latent variables in all models, which are all Gaussian except for JointVAE where one of them is replaced with a categorical variable. As Gaussian variables are parameterized by mean and standard deviation while the categorical variables are parameterized by logits, the final feature dimensionality is 12 for the models except for JointVAE where the dimensionality is 13 for DSPRITE and 14 for 3DSHAPES.

| Encoder | Decoder |
| --- | --- |
| Conv4x4s2p1, 32 channels | FC, 256 features |
| Conv4x4s2p1, 32 channels | FC, 64x4x4 features |
| Conv4x4s2p1, 64 channels | ConvT4x4s2p1, 64 channels |
| Conv4x4s2p1, 64 channels | ConvT4x4s2p1, 32 channels |
| FC, 256 features | ConvT4x4s2p1, 32 channels |
| FC, * features | ConvT4x4s2p1, 1 or 3 channels |

Recall that, in the toy model, the representation $\mathbf{z}$ is drawn from $\mathcal{N}(\mathbf{y}, \sigma^2\mathbf{I})$, where the generative factors $\mathbf{y}$ follow the standard Gaussian distribution. Therefore, the marginal distribution of the representation is $p(\mathbf{z}) = \mathcal{N}(\mathbf{z}; \mathbf{0}, (1+\sigma^2)\mathbf{I})$. When the representation is conditioned by a single factor $y_k$, it follows $p(\mathbf{z}|y_k) = \mathcal{N}(\mathbf{z}; y_k\mathbf{e}_k, (1+\sigma^2)\mathbf{I} - \mathbf{e}_k\mathbf{e}_k^\top)$. By substituting the covariance matrices of these distributions to $\Sigma$ in Eq.15 and computing their differences, we obtain the mutual information terms as follows.

$$I(y_k; \tilde{z}_\ell^{\text{red}}) = \begin{cases} \frac{1}{2}\log\frac{1+\sigma^2}{\sigma^2} & (k=\ell), \\ 0 & (k\neq\ell), \end{cases} \quad I(y_k; \tilde{z}_{L+\ell}^{\text{red}}) = \begin{cases} \frac{1}{2}\log\frac{1+\alpha^2(1+\sigma^2)}{1+\alpha^2(1+\sigma^2-(1-\frac{2}{K})^2)} & (k=\ell), \\ \frac{1}{2}\log\frac{1+\alpha^2(1+\sigma^2)}{1+\alpha^2(1+\sigma^2-\frac{4}{K^2})} & (k\neq\ell), \end{cases}$$

$$I(y_k; \tilde{\mathbf{z}}_{\backslash\ell}^{\text{red}}) = \begin{cases} \frac{1}{2}\log\frac{1+\alpha^2(1+\sigma^2)}{1+\alpha^2\sigma^2} & (k=\ell), \\ \frac{1}{2}\log\frac{1+\sigma^2}{\sigma^2} & (k\neq\ell), \end{cases} \quad I(y_k; \tilde{\mathbf{z}}_{\backslash L+\ell}^{\text{red}}) = \frac{1}{2}\log\frac{1+\sigma^2}{\sigma^2},$$

$$I(y_k; \tilde{z}_\ell^{\text{syn}}) = \begin{cases} \frac{1}{2}\log\frac{1+\sigma^2+\alpha^2}{\sigma^2+\alpha^2} & (k=\ell), \\ 0 & (k\neq\ell), \end{cases} \quad I(y_k; \tilde{z}_{L+\ell}^{\text{syn}}) = 0, \qquad (16)$$

$$I(y_k; \tilde{\mathbf{z}}_{\backslash\ell}^{\text{syn}}) = \begin{cases} 0 & (k=\ell), \\ \frac{1}{2}\log\frac{1+\sigma^2}{\sigma^2} & (k\neq\ell), \end{cases} \quad I(y_k; \tilde{\mathbf{z}}_{\backslash L+\ell}^{\text{syn}}) = \begin{cases} \frac{1}{2}\log\frac{(1+\sigma^2)(1+\sigma^2+\alpha^2)}{\sigma^2(1+\sigma^2+\alpha^2)+\alpha^2(1-\frac{2}{K})^2} & (k=\ell), \\ \frac{1}{2}\log\frac{(1+\sigma^2)(1+\sigma^2+\alpha^2)}{\sigma^2(1+\sigma^2+\alpha^2)+\alpha^2\frac{4}{K^2}} & (k\neq\ell) \end{cases}$$

$$I(y_k; \tilde{\mathbf{z}}^{\text{red}}) = I(y_k; \tilde{\mathbf{z}}^{\text{syn}}) = \frac{1}{2}\log\frac{1+\sigma^2}{\sigma^2}.$$

The metrics in Table 1 are computed from these quantities. In addition, we can compute other partial information terms as follows.

$$\mathcal{R}(y_k; \tilde{z}_k^{\text{red}}, \tilde{\mathbf{z}}_{\backslash k}^{\text{red}}) = \frac{1}{2}\log\frac{1+\alpha^2(1+\sigma^2)}{1+\alpha^2\sigma^2}, \quad \mathcal{C}(y_k; \tilde{z}_k^{\text{red}}, \tilde{\mathbf{z}}_{\backslash k}^{\text{red}}) = 0,$$

$$\mathcal{R}(y_k; \tilde{z}_k^{\text{syn}}, \tilde{\mathbf{z}}_{\backslash k}^{\text{syn}}) = 0, \qquad\qquad \mathcal{C}(y_k; \tilde{z}_k^{\text{syn}}, \tilde{\mathbf{z}}_{\backslash k}^{\text{syn}}) = \frac{1}{2}\log\frac{(1+\sigma^2)(\sigma^2+\alpha^2)}{\sigma^2(1+\sigma^2+\alpha^2)}.$$

We can observe that the redundancy and synergy terms effectively increase with the corresponding attacks.

## D MODEL ARCHITECTURES AND TRAINING HYPERPARAMETERS

The architectures of the encoder and decoder used in the experiments are listed in Table 2. We used ReLU nonlinearity at each convolutional layer except for the final output of each network. In addition, we used for the critic in FactorVAE a feedforward network with five hidden layers each of which consists of 1,000 leaky ReLU units with the slope coefficient of 0.2.

Table 3: Training hyperparameters used for DSPRITES experiments.

| Model | $\beta$-VAE | FactorVAE | JointVAE | $\beta$-TCVAE |
|---|---|---|---|---|
| Batch size | 64 | 64 | 64 | 2,048 |
| Iterations | 300,000 | 300,000 | 300,000 | 30,000 |
| Adam $\alpha$ | $5 \times 10^{-4}$ | $1 \times 10^{-4}$ | $5 \times 10^{-4}$ | $1 \times 10^{-3}$ |
| Adam $\beta_1, \beta_2$ | 0.9, 0.999 | 0.9, 0.999 | 0.9, 0.999 | 0.9, 0.999 |
| Critic Adam $\alpha$ | - | $1 \times 10^{-4}$ | - | - |
| Critic Adam $\beta_1, \beta_2$ | - | 0.5, 0.9 | - | - |
| Discrete variable capacity $C_c$ | - | - | 1.1 | - |
| Continuous variable capacity $C_z$ | - | - | 40 | - |
| Regularization coefficient | $\beta = 4$ | $\gamma = 35$ | $\gamma = 150$ | $\beta = 6$ |

Table 4: Training hyperparameters used for 3DSHAPES experiments.

| Model | $\beta$-VAE | FactorVAE | JointVAE | $\beta$-TCVAE |
|---|---|---|---|---|
| Batch size | 64 | 64 | 64 | 2,048 |
| Iterations | 500,000 | 500,000 | 500,000 | 50,000 |
| Adam $\alpha$ | $1 \times 10^{-4}$ | $1 \times 10^{-4}$ | $1 \times 10^{-4}$ | $1 \times 10^{-3}$ |
| Adam $\beta_1, \beta_2$ | 0.9, 0.999 | 0.9, 0.999 | 0.9, 0.999 | 0.9, 0.999 |
| Critic Adam $\alpha$ | - | $1 \times 10^{-5}$ | - | - |
| Critic Adam $\beta_1, \beta_2$ | - | 0.5, 0.9 | - | - |
| Discrete variable capacity $C_c$ | - | - | 1.1 | - |
| Continuous variable capacity $C_z$ | - | - | 40 | - |
| Regularization coefficient | $\beta = 4$ | $\gamma = 20$ | $\gamma = 150$ | $\beta = 4$ |

The hyperparameters used in training are listed in Table 3 and Table 4. For hyperparameters not listed in the tables, we used the values suggested in the original papers.

## E FULL RESULTS FOR FACTOR-WISE PID ANALYSES

We illustrate the estimated bounds of PID terms for each factor in Figure 6, whose small version appeared in Figure 4. We summarize these results in Table 5. This table is made by observing and categorizing the plots in Figure 6 into some patterns as follows. Note that we ignore the error bars here.

1. If the lower bound of the unique information is larger than the upper bounds of the redundant and complementary information, mark the plot as *disentangled*. In this case, the model is determined as successfully disentangling the factor regardless of the concrete definition of PID. Note that the model does not necessarily learn the factor completely; see the figure for how much the information of the factor is uniquely captured by a latent variable.

2. If the upper and lower bounds of the redundant information are larger than those of the complementary information by a margin, mark the plot as *redundant*. In this case, the model entangles the factor in a redundant way. As we analyzed in Section 5.2, this indicates that the latent variables are highly dependent.

3. If the upper and lower bounds of the complementary information are larger than those of the redundant information by a margin, mark the plot as *synergistic*. In this case, the model entangles the factor in a synergistic way. This may occur even if the latent variables are independent. We may require additional inductive biases to help the model disentangle the factor.

4. Otherwise, mark the plot as *flat*.

## F POSSIBLE EXPLANATIONS FOR HIGH REDUNDANCY IN JOINTVAE

We observed in Table 5 and Figure 6 that JointVAE suffers from high redundancy in all the factors. To explain this phenomenon, we review the training scheme of JointVAE (Dupont, 2018). The

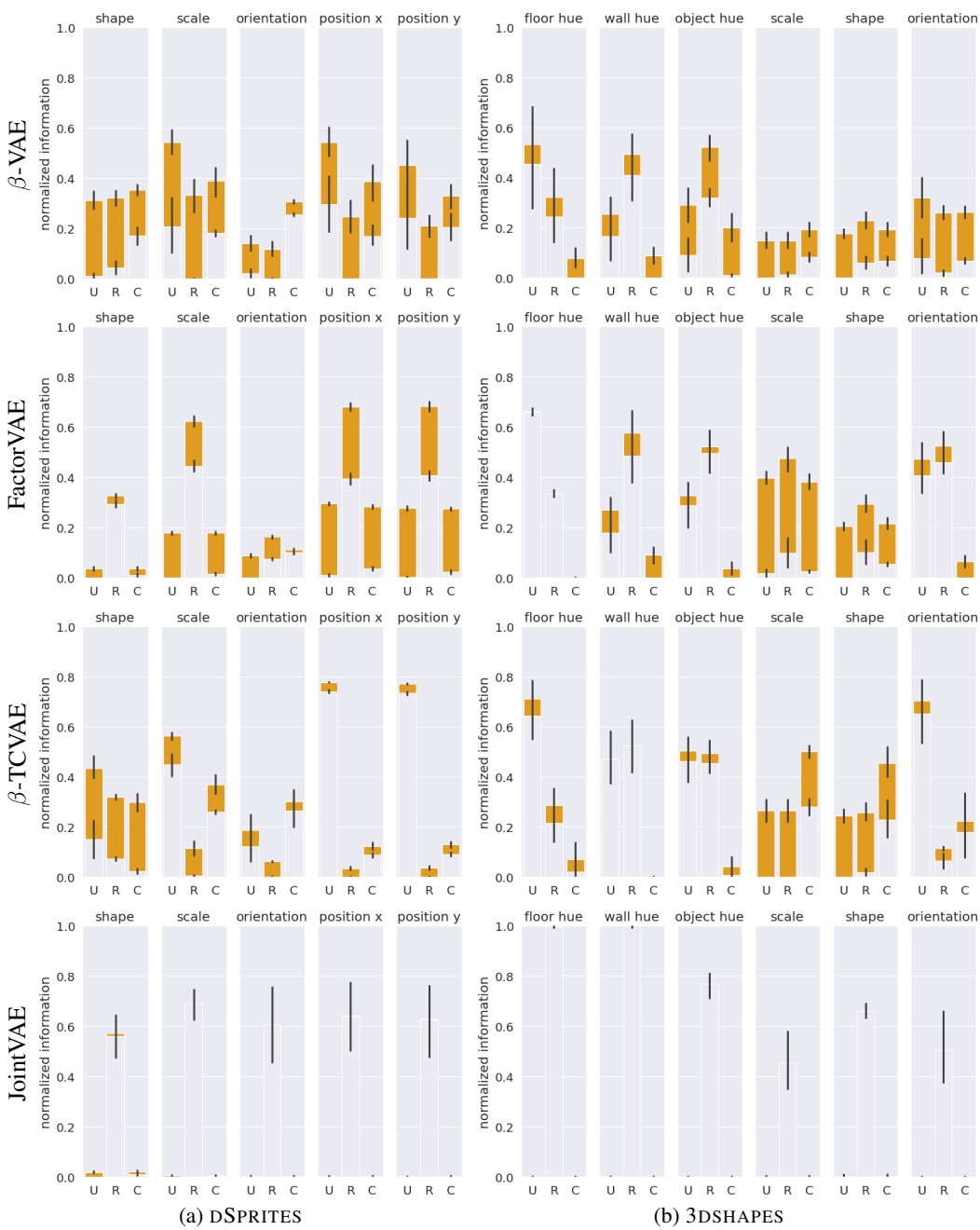

Figure 6: Large version of Figure 4.

Table 5: Qualitative summary of PID decomposition for model-factor pairs. Each pair is explained by the following terms: **disentangled**: the unique information is larger than other terms; **synergistic**, **redundant**: the corresponding PID term is large; **flat**: no term exceeds others much, which indicates that all three terms are small (i.e., multiple variables contain distinct information of the factor) or both redundancy and synergy are large. Note that these qualitative analyses are only applicable to our experimental settings. In particular, high redundancy of JointVAE (marked by $*$ in the table) may be caused by a capacity hyperparameter. See Appendix G for details.

| Dataset | Factor | $\beta$-VAE | FactorVAE | $\beta$-TCVAE | JointVAE |
|---|---|---|---|---|---|
| DSPRITES | shape | flat | redundant | flat | redundant* |
| | scale | synergistic | redundant | **disentangled** | redundant* |
| | orientation | synergistic | flat | synergistic | redundant* |
| | position x | synergistic | redundant | **disentangled** | redundant* |
| | position y | synergistic | redundant | **disentangled** | redundant* |
| 3DSHAPES | floor hue | **disentangled** | **disentangled** | **disentangled** | redundant* |
| | wall hue | redundant | redundant | redundant | redundant* |
| | object hue | redundant | redundant | redundant | redundant* |
| | scale | flat | flat | synergistic | redundant* |
| | shape | flat | flat | synergistic | redundant* |
| | orientation | flat | redundant | **disentangled** | redundant* |

Table 6: KL terms and total correlation of latent variables learned by JointVAE. The values in parentheses show the standard deviation.

| Dataset | $D_{\mathrm{KL}}(p(z_1|x)\|U(z_1))$ | $D_{\mathrm{KL}}(p(\mathbf{z}_{2:L}|x)\|\mathcal{N}(\mathbf{z}_{2:L};\mathbf{0},\mathbf{I}))$ | Total correlation of $\mathbf{z}$ |
|---|---|---|---|
| DSPRITES | 1.10 ($\pm$0.00) | 40.00 ($\pm$0.04) | 25.02 ($\pm$0.47) |
| 3DSHAPES | 1.10 ($\pm$0.00) | 39.99 ($\pm$0.05) | 25.98 ($\pm$0.56) |

JointVAE model consists of one categorical variable $z_1$ and $L-1$ Gaussian variables $\mathbf{z}_{2:L}$. Let $U(z_1)$ be the uniform categorical distribution and $p_d(x|\mathbf{z})$ be the decoder to be learned simultaneously. Then, the objective function of JointVAE is

$$\mathcal{L} = \mathbb{E}_{p(\mathbf{z}|x)}[\log p_d(x|\mathbf{z})] - \gamma|D_{\mathrm{KL}}(p(z_1|x)\|U(z_1)) - c_1| - \gamma|D_{\mathrm{KL}}(p(\mathbf{z}_{2:L}|x)\|\mathcal{N}(\mathbf{z}_{2:L};\mathbf{0},\mathbf{I})) - c_2|,$$

which involves three hyperparameters: the regularization coefficient $\gamma$, the capacity of the discrete variable $c_1$, and the capacity of continuous variables $c_2$. Throughout training, $\gamma$ is kept constant, while the capacities $c_1$ and $c_2$ are gradually increased and saturated at the predefined maximum values ($C_c$ and $C_z$, respectively) in the middle of training. As the capacities are positive at the end of training, this indicates that the KL terms, which include the total correlation of the latent variables (Kim & Mnih, 2018; Chen et al., 2018), are large in the trained model. Therefore, as we discussed in Section 5.2, this model does not add pressure on the representation to be less redundant, which may explain the high redundancy[3]. See Appendix G for the results with varying $C_z$, whose results also support the above hypothesis as low capacities induce lower redundancy. This training scheme is designed to align the amount of information captured by discrete and continuous variables, which seem to be successful as we observed, i.e., it effectively captures the shape factor in both datasets compared to the other methods.

## G  INFLUENCE OF REGULARIZATION HYPERPARAMETERS ON DISENTANGLEMENT

We trained each model with varying hyperparameters ($\beta$ for $\beta$-VAE and $\beta$-TCVAE, $\gamma$ for FactorVAE, and the capcity of continuous variables $C_z$ for JointVAE) and investigated how the metrics as well as PID terms are affected by the regularization strengths[4].

---

[3]To confirm that large capacity actually causes large dependency, we measured the KL terms and the total correlation of the trained JointVAE models. We summarize the results in Table 6. These values indicate that the KL terms are actually close to the capacity hyperparameters (see Table 3 and Table 4), and more than a half of them are occupied by the total correlation.

[4]For JointVAE, we chose the final capacity of continuous variables instead of the regularization coefficient as we expect the capacity to be more relevant to disentanglement. The former controls the KL divergence

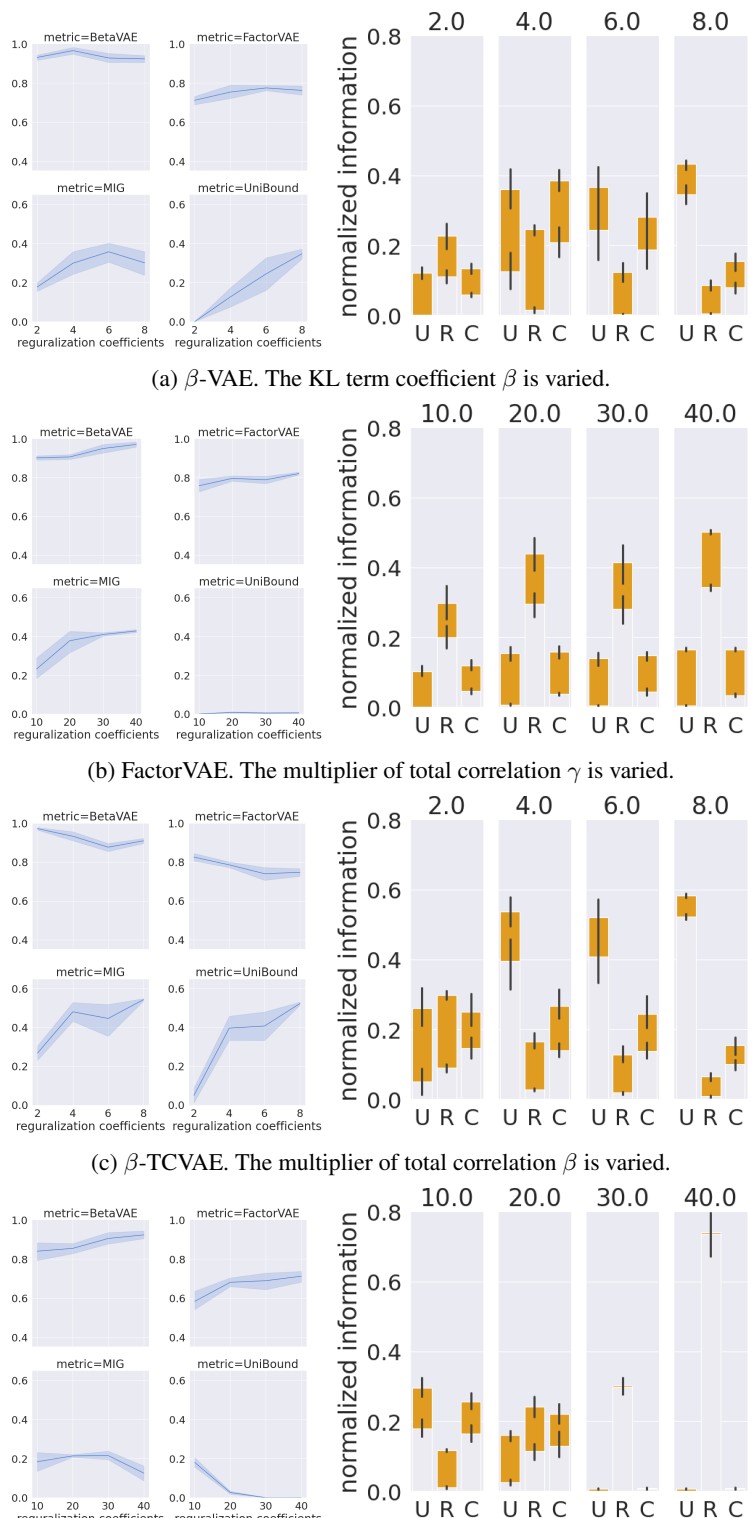

(a) $\beta$-VAE. The KL term coefficient $\beta$ is varied.

(b) FactorVAE. The multiplier of total correlation $\gamma$ is varied.

(c) $\beta$-TCVAE. The multiplier of total correlation $\beta$ is varied.

(d) JointVAE. The final capacity of continuous variables $C_z$ is varied. Note that higher capacity induces lower regularization effect.

Figure 7: Metrics and PID estimations for varying regularization coefficients on DSPRITES. For each model, the left panel shows how each metric reacts to changing the hyperparameter. The right panel shows the PID esitmations of resulting representations. In all plots, the horizontal axis corresponds to the regularization hyperparameter.

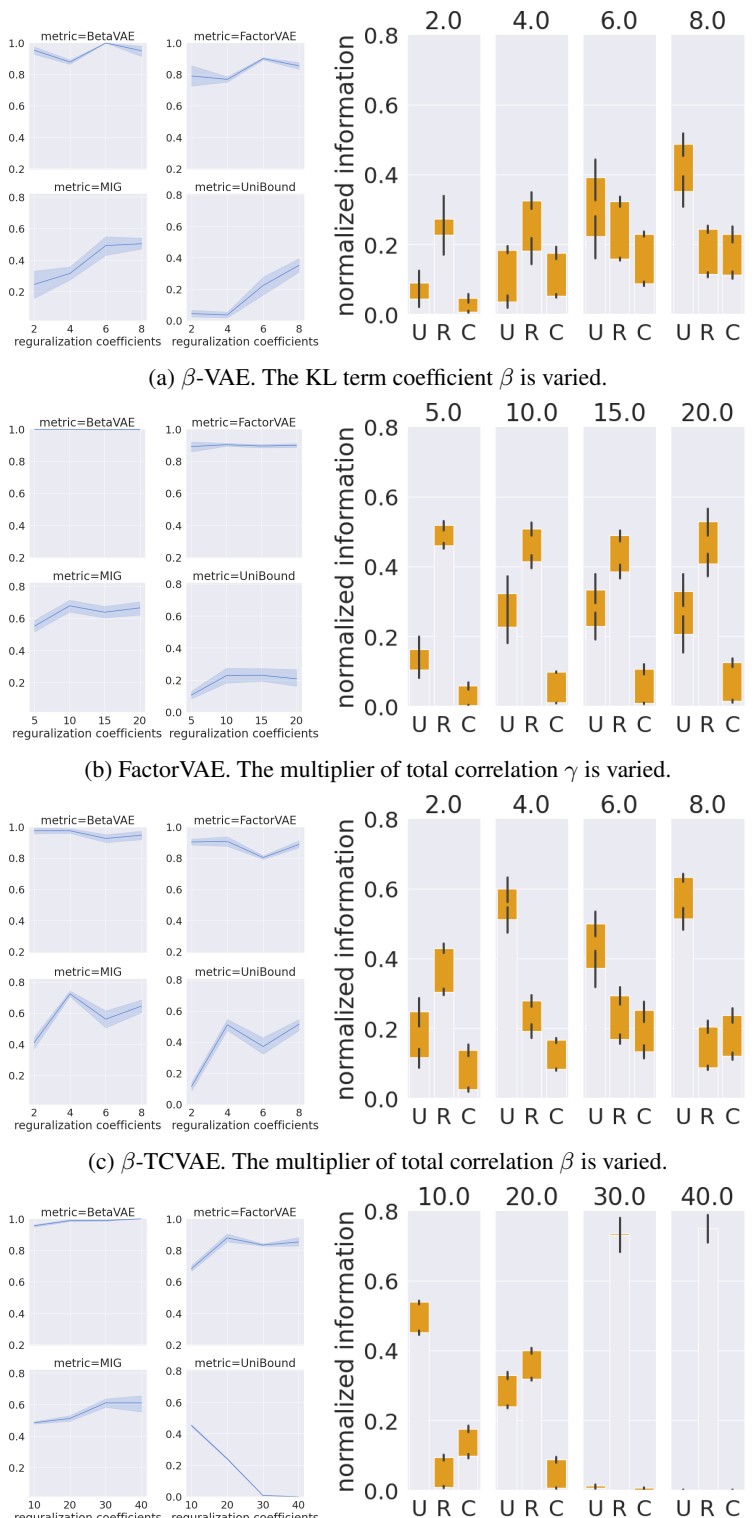

(a) $\beta$-VAE. The KL term coefficient $\beta$ is varied.

(b) FactorVAE. The multiplier of total correlation $\gamma$ is varied.

(c) $\beta$-TCVAE. The multiplier of total correlation $\beta$ is varied.

(d) JointVAE. The final capacity of continuous variables $C_z$ is varied. Note that higher capacity induces lower regularization effect.

Figure 8: Metrics and PID estimations for varying regularization coefficients on 3DSHAPES.

The results for DSPRITES and 3DSHAPES are illustrated in Fig.7 and Fig.8, respectively. From the left panels, we can observe that the UNIBOUND metric is positively correlated with the regularization strength[5]. It indicates that the regularization method introduced by each model positively contributes to disentanglement in the PID perspective. We also estimated PID bounds in the right panels. They show some interesting effects of regularization against the types of entanglement. For example, in JointVAE, the representation has low redundancy when the capacity of continuous variables is small, and the redundancy grows significantly when we increase the capacity. It indicates that a large capacity makes the model enforce each latent variable to capture details of the input images, ignoring how the information is redundantly captured by other variables.

## H    LARGE FIGURES OF EXPERIMENTAL RESULTS

We put large versions of Figure 3 and Figure 4 in Figure 9 and Figure 6, respectively, for finer rendering.

## I    TRAINING AND EVALUATION STABILITY

We illustrate the disentanglement scores of models trained with eight training seeds in Figure 10 and Figure 11. As each disentanglement metric involves sampling during evaluation, the evaluated score has some randomness even if we fix the training random seed. This plot reveals that the deviation caused by randomness in evaluating disentanglement metrics is much smaller than the deviation caused by randomness in training each model.

---

between the aggregated posterior of continuous variables and their prior at the end of training, while the latter controls the strength of enforcing the KL divergence to be close to the capacity. See the original paper (Dupont, 2018) for more details.

[5]In JointVAE, the hyperparameter controls the final KL term; hence, a smaller hyperparameter should induce more disentangled representation.

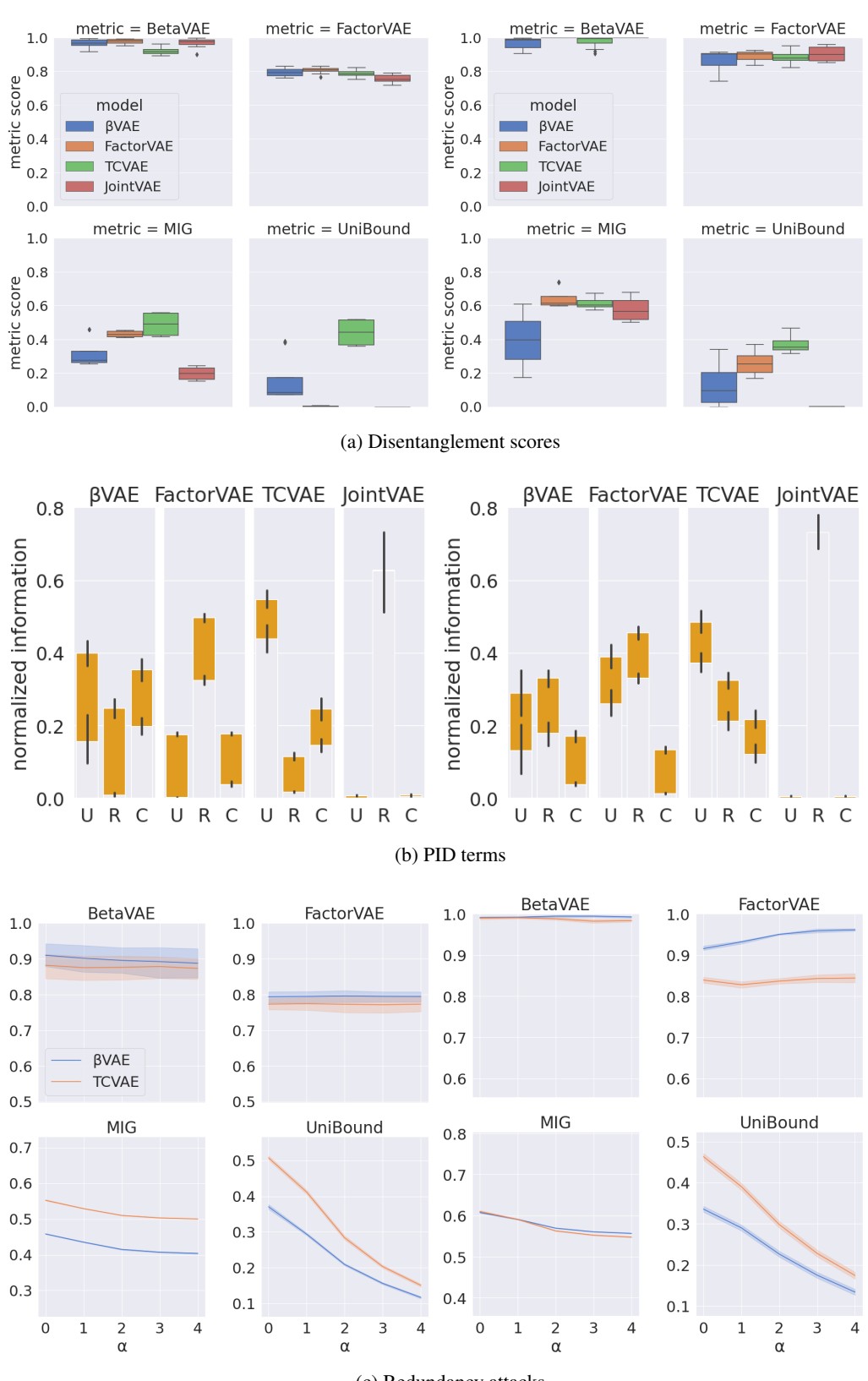

Figure 9: Large versions of Fig.3. (Left) Results for DSPRITES. (Right) Results for 3DSHAPES.

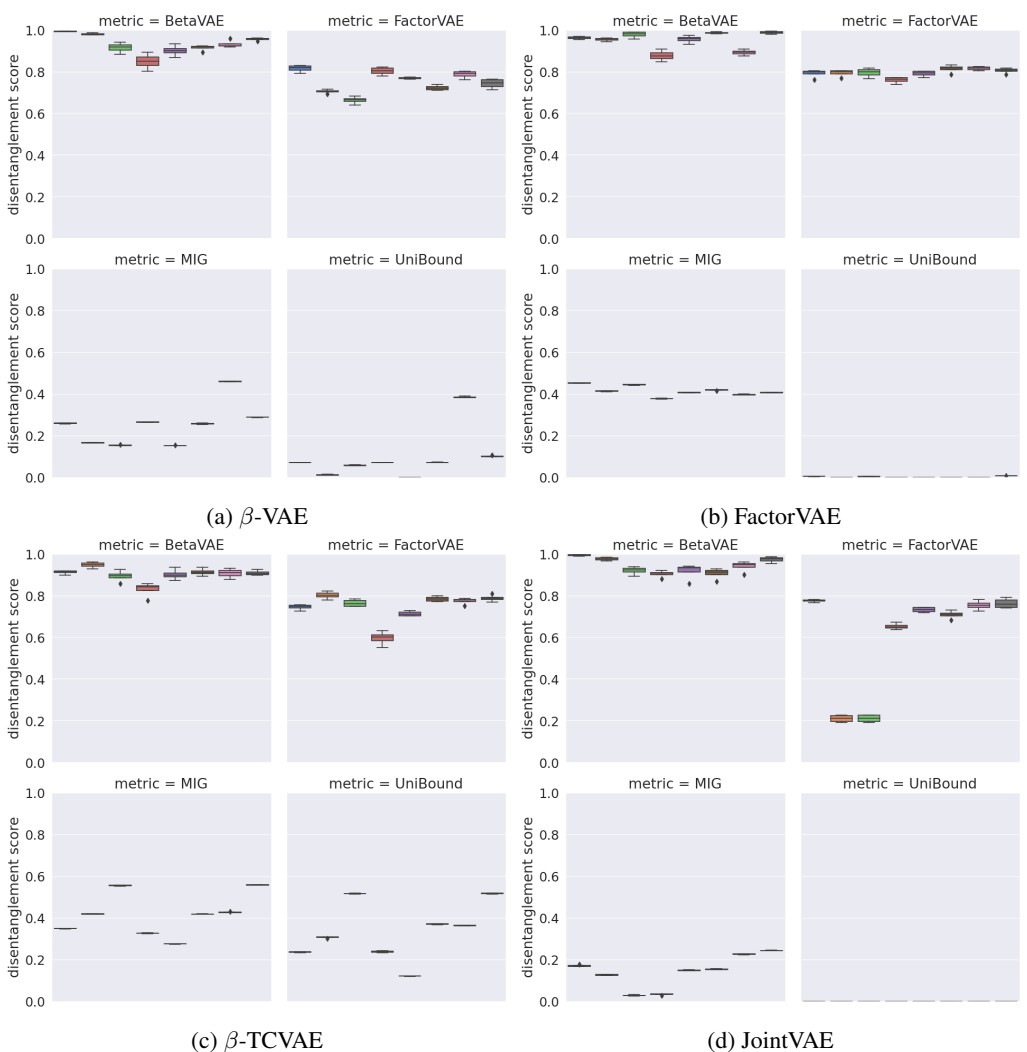

Figure 10: Disentanglement scores before aggregating across different training seeds for DSPRITES. We optimized parameters for each model eight times with different random seeds, whose scores are illustrated by the eight boxes in each plot.

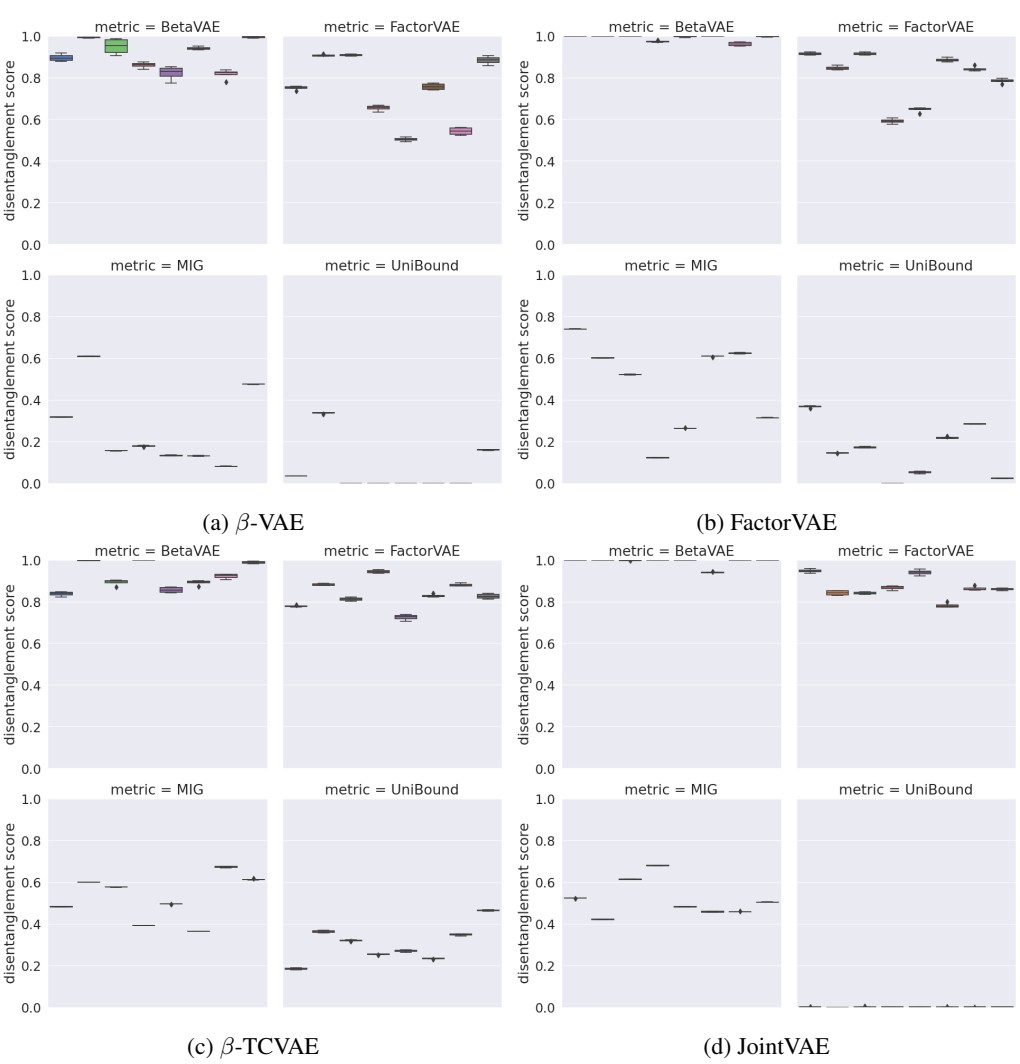

Figure 11: Disentanglement scores before aggregating across different training seeds for 3DSHAPES.

