# OpenReview forum: "Disentanglement Analysis with Partial Information Decomposition"
_ICLR.cc/2022/Conference — ICLR 2022 Poster_

### Official Review · Reviewer_kftq · 2021-10-31

**Correctness:** 3
**Technical Novelty And Significance:** 2
**Empirical Novelty And Significance:** 2
**Recommendation:** 6
**Confidence:** 4

**Main Review:**

[Strengths]
- Applying PID for disentanglement analysis looks quite interesting, and the derivation generally looks thorough.
- The paper proposes a new disentanglement metric to address the weakness of existing metrics, which do not capture multi-variable interactions properly.
- The paper is well backed-up by a comprehensive supplementary material that seems to contain all the details required for reproducibility. The paper is fairly clearly written and easy to follow.

[Comments]
- It would be better to quantitatively and/or visually support the claim “Current metrics, however, may fail to detect entanglement that involves more than two variables, e.g., representations that duplicate and rotate generative factors in high dimensional spaces,” which is the most important motivation of this work.
- Though simulating some entanglement attacks is worth trying, I am not sure whether these attacks actually and frequently happen in disentanglement learning. It would be better to find and show the cases similar with the redundancy and synergy attacks by analyzing learned features.
- Unsupervised disentanglement learning with VAEs has been known to provide noisy results that are extremely affected by random seeds, architectures, and hyper-parameters (Locatello, 2019). Because the experiments merely relied on unsupervised VAEs with a single fixed architecture and few hyperparameter settings, I am not sure whether the results obtained under such a limited setup are reliable and whether this work could attract the attention of readers in the disentanglement learning field. For example, is the trend of the results in Figure 3 consistent for other architectures and settings? How did the authors set the regularization coefficients for disentanglement learning (beta, gamma) and other hyperparameters? Furthermore, I think drawing a conclusion described in Table 1 may be dangerous because the examined VAEs were tested under a very limited setup. I would suggest the authors to include (i) additional results under various settings and/or (ii) some results using semi- or weakly-supervised disentanglement models (Locatello, 2020a; Locatello, 2020b; Chen, 2020; Feng, 2018; Szabó, 2018; Kingma, 2014).
-I am not sure whether enforcing nonnegativity in the RHS of eq. (5) using the max{0,*} operation is theoretically allowed. What happens if the max{0,*} operation is removed in the UniBound metric?
- The results of JointVAE using the proposed metric seem weird. Though I checked the authors’ claims (This can be viewed as the effect of introducing a discrete variable into the representation; The high redundancy in JointVAE can also be explained by the lack of independence), they were unclear for me. In particular, the claim “As the capacities are positive at the end of training, this indicates that the KL terms, which include the total correlation of the latent variables (Kim & Mnih, 2018; Chen et al., 2018), are large in the trained model” may be incorrect because the KL term includes not only the TC term but also the data-latent MI and dimension-wise KL terms, meaning that a large KL does not directly indicate a large TC. It would be better to add more clear explanations (e.g., by actually measuring the TC terms). Furthermore, it would be helpful to see whether these results are also observed with other discrete-variable VAEs such as CascadeVAE (Jeong, 2019).
- It would be better to include some qualitative results (latent traversal, t-SNE embedding space visualization) for visually understanding how feature spaces change according to the value of the metrics.
- Abstract: asses -> assess

(Locatello, 2019) Challenging Common Assumptions in the Unsupervised Learning of Disentangled Representations, ICML'19

(Locatello, 2020a) Weakly-supervised disentanglement without compromises, ICML'20

(Locatello, 2020b) Disentangling factors of variations using few labels, ICLR'20

(Chen, 2020) Weakly supervised disentanglement by pairwise similarities, AAAI'20

(Feng, 2018) Dual Swap Disentangling, NeurIPS’18

(Szabó, 2018) Challenges in Disentangling Independent Factors of Variation, ICLRW'18

(Kingma, 2014) Semi-supervised Learning with Deep Generative Models, NeurIPS’14

(Jeong, 2019) Learning Discrete and Continuous Factors of Data via Alternating Disentanglement, ICML’19


**Summary Of The Paper:**

The authors leveraged partial information decomposition (PID) for analyzing multi-variable interactions in latent representations. The PID framework shows that (i) the mutual information between a latent variable and a generative factor can be divided into unique, redundant, and synergistic information terms, and (ii) the uniqueness term corresponds to the degree of disentanglement. The authors also introduced a disentanglement metric by modifying MIG and conducted experiments with VAE-based models on two datasets.

**Summary Of The Review:**

The main idea (applying PID for disentanglement analysis) looks interesting, and the underlying technical contributions seem good. However, the empirical results are very weak and not impressive. I thus find it difficult to argue for acceptance of the work.

---

> ### Author Response · Authors · 2021-11-19
> **Response to Reviewer kftq (1/2)**
>
> Thank you for the review. We appreciate your detailed comments with helpful suggestions.
>
>
> > Unsupervised disentanglement learning with VAEs has been known to provide noisy results that are extremely affected by random seeds, architectures, and hyper-parameters (Locatello, 2019) ...
>
> At least for random seeds, we trained models many times changing the seed as we stated in Sec.5.2. So far, the result is affected by the randomness, though the deviation is small enough to interpret the results as we showed through figures (note that the error bars in Fig.3ab, 5, and 6 are mostly due to the training randomness).
>
> As it is interesting to see how the choice of hyperparameters affect the results, we added experiments on varying regularization coefficients in Appendix G. The results behave consistently (i.e., disentanglement scores consistently react to the increasingly regularized or disregularized models), which should back up the reliability of our discussions. Thank you for raising this point!
>
> On concerns on Table 1, we have written additional notes that it may be affected by experimental settings (in particular on the results for JointVAE which we observed in Appendix G that the regularization hyperparameter affects the results) and moved the table to the appendix (Table 5). We have replaced the results in the main text with the quantitative plots, which was requested by Reviewer mDLP.
>
>
> > I am not sure whether these attacks actually and frequently happen in disentanglement learning.
>
> We intended with these attacks to control the amount of redundancy or synergy in a representation through a scalar parameter $\alpha$ to see how different metrics react to the changing representation. Controlling redundancy or synergy in such a way is hard for learned representations. In machine learning, we often verify the validity of a method using controllable synthetic data.
>
> Although we do not directly relate learned representations to the attacks themselves, there are cases of learned representations with high redundancy and synergy. For example, the different values between MIG and UniBound in FactorVAE and JointVAE (see Fig.3a) indicate that these models actually encode individual factors redundantly among multiple latent variables.
>
> If the name “attack” sounds misleading to you, we are ready to change the naming, e.g. to “simulated entanglement injection.” We would like to hear your opinion.
>
>
> > I am not sure whether enforcing nonnegativity in the RHS of eq. (5) using the max{0,} operation is theoretically allowed.
>
> We clipped the values because we understand these terms as lower bounds of PID terms, and negative bounds are vacuous as PID terms are nonnegative by design. We used the results to analyze PID terms through bounds rather than to analyze bounds themselves. In some cases, we may give meaning to bounds themselves, as we did for the lower bound of redundant information; it may be meaningful to compare the values before clipping in such a case. We did not do such analysis in the paper as that is not our main motivation.
>
>
> > It would be better to quantitatively and/or visually support the claim …
> > …
> > It would be better to include some qualitative results (latent traversal, t-SNE embedding space visualization) for visually understanding how feature spaces change according to the value of the metrics.
>
> We do not conduct visualization analysis like latent traversal because it is not suitable for capturing multi-variable interactions. For example, in latent traversal, one varies only one latent variable at a time. It is a fundamental limitation of this visualization method, as we cannot place images at various points in multidimensional latent space to a two-dimensional figure. Since we are interested in how multiple variables are related, this limitation is critical for this work.
>
> Multidimensional entanglement may not be found through visualization. We argue that understanding of such a multidimensional phenomenon should be driven by quantitative measurements rather than qualitative analysis of visualization. We implemented such quantitative measurements through this work and confirmed the existence of multidimensional entanglement through experimental results (e.g., the differences between MIG and UniBound in FactorVAE and JointVAE in Fig.3a are signals of the existence).

---

> > ### Author Response · Authors · 2021-11-19
> > **Response to Reviewer kftq (2/2)**
> >
> > > The results of JointVAE using the proposed metric seem weird. Though I checked the authors’ claims (This can be viewed as the effect of introducing a discrete variable into the representation; The high redundancy in JointVAE can also be explained by the lack of independence), they were unclear for me. …
> >
> > We measured the KL terms and the total correlation of JointVAE and summarized in Table 6 in Appendix F. We confirmed that (1) the KL terms are actually close to the capacity hyperparameters, and (2) more than a half of the KL values are occupied by the total correlation. This result supports our claim on how the capacity hyperparameters of JointVAE affect the dependency of latent variables.
> >
> >
> > > Abstract: asses -> assess
> >
> > We fixed it.

---

> > > ### Comment · Reviewer_kftq · 2021-11-21
> > > **Post-rebuttal review**
> > >
> > > I have carefully read the rebuttal, and I would like to raise my rating to 6 for the following reasons:
> > > - The authors clearly resolved the concerns regarding experiments by adding the results on multiple regularization coefficients and the analysis on the KL and TC terms of JointVAE. They also improved the presentation quality with additional notes and re-organized some results according to the reviewers’ comments for a better understanding.
> > > - I appreciate the authors pointing out the existence of redundant and synergistic information in Figure 3 which cannot be well captured by MIG. I think using the term “attack” is fine (instead of “simulated entanglement injection”) because the two indicate exactly the same thing, while the latter feels quite lengthy.
> > > - I also agree with the authors’ opinion on the clipping and visualization. The PID terms are nonnegative by construction and comparing whether to use clipping or not is not the primary focus of this paper. Furthermore, visualizing multi-variable entanglement may be difficult, and analyzing it with quantitative analysis may be more meaningful.
> > >
> > > I believe studying the multi-variable interaction is important in the field of disentanglement learning (as noted by other reviewers), the proposed approach based on PID is quite novel and interesting, and thus the paper could be a good fit for the conference.

---

> > > > ### Author Response · Authors · 2021-11-21
> > > > **Re: Post-rebuttal review**
> > > >
> > > > Thank you for evaluating the revision and our comments!

---

### Official Review · Reviewer_ezka · 2021-11-01

**Correctness:** 3
**Technical Novelty And Significance:** 2
**Empirical Novelty And Significance:** 2
**Recommendation:** 5
**Confidence:** 5

**Main Review:**

Pros:
1) The problem that the authors want to address is important.
2) The proposed metric is reasonable.

Cons:
1) The paper misses important references and discussion of prior work.
2) The presentation is not clear.
3) Experiments are limited to only 2 toy datasets and do not really highlight the advantage of the proposed metric.
4) The proposed metric seems not practical compared to existing metrics.


**Summary Of The Paper:**

This paper proposes a metric for evaluating disentanglement inspired by Partial Information Decomposition (PID). It also proposes 2 “entanglement” attacks to highlight the advantages of the proposed metric.

**Summary Of The Review:**

1) Novelty:
- The idea of an information-theoretic analysis framework for disentangled representations was proposed by Do & Tran [1] but is not discussed in the paper.
- The metric UniBound proposed in the paper has a very similar formula to those of the WSEPIN and WINDIN metrics in [1] but with the input image x replaced by the label y. I think the authors should discuss this similarity in their paper.
- The authors should also discuss and compare their metrics with other up-to-date improvements of MIG such as JEMMIG [1] or MIG-sup [2]. Please check [3] for more detail of these metrics.

2) Clarity in the presentation:
- I think the authors abuse notations in their paper which causes difficulty in reading. For example, if $\mathcal{R}$, $\mathcal{U}$, $\mathcal{C}$, $\mathcal{I}$ are just mutual information, the authors should write all of them as $I$. I think the variables inside these terms are enough to define their meanings.
- I don’t understand the meaning of the symbol $\backslash$ in the mutual information. Is it equivalent to the symbol $\|$ for conditional probability? If they are equivalent, I think the authors should write $\|$ to make the mutual information more familiar to readers. Otherwise, they need to write explicit formulas of $\mathcal{U}(y_k; z_{\ell} \backslash z_{\ell '})$ and $\mathcal{U}(y_k; z_{\ell '} \backslash z_{\ell})$ since I cannot guess what they are.
- The derivation of the proposed metric UniBound in Eq. 5 only uses Unique Information ($\mathcal{U}$) and neither Redundant Information ($\mathcal{R}$) nor Synergetic Information ($\mathcal{C}$). I wonder how the authors can relate their metric to “redundancy” and “synergy”?
- How the lower bound and upper bound of each term are related to disentanglement is not well discussed. I am almost unable to deduce anything when looking at Fig. 3b.

3) Correctness and practicality of the method:
- I don’t really understand why the latent variable $z$ is concatenated with a new vector in case of Redundancy and Synergy attacks? Doesn’t it double the length of $z$ and make $z$ no longer a suitable input for the decoder?
- I am concerned about the practicality of the proposed metric as it is computed by summing over many small probability density values ($\log\sum_{x} p(z|x)$ in Eq. 8), which often leads to numerical instability if $z$ is high-dimensional. This is serious because incorrect results can cause incorrect interpretations. MIG and its improvements (JEMMIG, MIG-sup), however, do not suffer from this problem since these metrics use the conditional probability of a single factor $p(z_{i}|x)$, not all factors. This problem was analyzed in [1] (Appdx. A5). I would like to hear the authors’ explanation about this. I also would like to know the dimensionality of the latent code z used in their experiments as I cannot see this value in Table 3, Appdx. C.
- Clipping $I(y_{k}, z_{\ell}) - I(y_{k}, z_{\backslash \ell})$ to be >= 0 makes the metric unable to compare between models that have $I(y_{k}, z_{\ell}) \ge I(y_{k}, z_{\backslash \ell})$. This is problematic because $I(y_{k}, z_{\backslash \ell})$ is likely to be larger than $I(y_{k}, z_{\ell})$. An example of this is the UniBound values for JointVAE in Fig. 3a.
- In Fig. 3a, UniBound produces very different results for FactorVAE on dSprite and on 3DShapes. I would like to hear the explanation for this.

[1] Theory and Evaluation Metrics for Learning Disentangled Representations, Do & Tran, ICLR 2020

[2] Progressive Learning and Disentanglement of Hierarchical Representations, Li et al., ICLR 2020

[3] Measuring Disentanglement - A Review of Metrics, Zaidi et al., 2021

---

> ### Author Response · Authors · 2021-11-19
> **Response to Reviewer ezka (1/2)**
>
> Thank you for the review. We appreciate your detailed comments on the technicalities and the understanding of the concepts.
>
>
> > The idea of an information-theoretic analysis framework for disentangled representations was proposed by Do & Tran [1] but is not discussed in the paper...
>
> Thank you for pointing this out. We added references to the main text and elaborated the comparisons in Appendix B.
>
> As you pointed out, WSEPIN has a similar interpretation as our framework: it can be seen as an upper bound of the unique information $\mathcal U(x; z_\ell \setminus \mathbf z_{\setminus\ell})$. It is still different from our metric in many aspects, e.g., (1) it is an optimistic evaluation in the PID perspective as it upper bounds the uniqueness while our UniBound is a pessimistic evaluation, and (2) it targets the information of $x$ in general rather than individual generative factors, with which we cannot analyze the information distribution for each factor as we did in Figure 4 and Table 5.
>
> RMIG extends MIG to imbalanced datasets. We think its main technique of inverting the conditioning of $y_k$ is also applicable to our framework. We conducted experiments on balanced datasets because whether the factors are balanced or not is not our main interest here, and hence used the original MIG.
>
> For JEMMIG, we are not very sure how its definition relates to our framework. It penalizes $H(z_\ell|y_k)$ more than (R)MIG, though this term already affects the score negatively in (R)MIG. Please see Appendix B for other metrics in the literature.
>
>
> > I think the authors abuse notations in their paper which causes difficulty in reading. For example, if $\mathcal R$, $\mathcal U$, $\mathcal C$, $\mathcal I$ are just mutual information, the authors should write all of them as $I$. I think the variables inside these terms are enough to define their meanings.
>
> We emphasize that these terms are NOT mutual information. The PID terms $\mathcal U$, $\mathcal R$, and $\mathcal C$ express the information distribution among more than two variables (namely $y_k$, $z_\ell$, and $\mathbf z_{\setminus\ell}$ in our case), while mutual information expresses the information between two variables. The information distribution among three variables are related with (but not fully determined by) mutual information through Eq.1-2. They are also illustrated in Fig.1 and Fig.2a.
>
> We do not provide exact formulae to define these terms because there have been several different definitions for these terms (Williams & Beer, 2010; Bertschinger et al., 2014; Finn & Lizier, 2018; 2020; Sigtermans, 2020) and we wanted to keep our analyses compatible with any of them. The PID framework was originally proposed by (Williams & Beer, 2010), where they established core conditions that the partial information terms should satisfy in general. As far as we know, our work is the first attempt to apply PID for disentanglement analysis, and a general discussion applicable to any particular definitions of PID is we think useful for later research, in which one may adopt any definitions in the literature.
>
> The interaction information is not mutual information as well, though it is exactly expressed by arithmetics of mutual information terms. We changed the notation from $\mathcal I$ to $I\!I$ in response to the comment by Reviewer 25MQ to avoid confusion.
>
>
> > I don’t understand the meaning of the symbol $\setminus$ in the mutual information.
>
> As we wrote in the footnote in page 2, this is a common notation in PID literature that emphasizes its asymmetricity and the resemblance to set difference operation. The unique information $\mathcal U(u; v_1 \setminus v_2)$ represents the information of $u$ contained in $v_1$ that is not conveyed by $v_2$. This can be equal to the mutual information $I(u; v_1)$, but is smaller in general. The governing equation of this term is $I(u; v_1) = \mathcal U(u; v_1 \setminus v_2) + \mathcal R(u; v_1, v_2)$, which reads "the information of $u$ is split into the part uniquely contained in $v_1$ against $v_2$ plus the part simultaneously represented by both $v_1$ and $v_2$". The resemblance to the set operator is best illustrated in the Venn-like diagram of Fig.1. Note that, in this diagram, three circles correspond to mutual information terms, while the separated four areas correspond to four PID terms.

---

> > ### Author Response · Authors · 2021-11-19
> > **Response to Reviewer ezka (2/2)**
> >
> > > The derivation of the proposed metric UniBound in Eq. 5 only uses Unique Information ($\mathcal U$) and neither Redundant Information ($\mathcal R$) nor Synergetic Information ($\mathcal C$). I wonder how the authors can relate their metric to “redundancy” and “synergy”?
> >
> > Excluding “redundancy” and “synergy” is a useful property of UniBound. We found that MIG includes redundancy, i.e., MIG can become large not due to large unique information but due to large redundant information, which is not suitable as a disentanglement metric. These points are illustrated in Fig.2b and Eq.4.
> >
> > It is correct that UniBound itself does not provide how the model entangles factors in terms of PID. We need other bounds to analyze such entanglement in details; please see entanglement analyses we did in Figure 3b.
> >
> >
> > > I don’t really understand why the latent variable $z$ is concatenated with a new vector in case of Redundancy and Synergy attacks? Doesn’t it double the length of $z$ and make $z$ no longer a suitable input for the decoder?
> >
> > We do not need to use the decoder for the evaluation with these attacks. We intended with these attacks to control the amount of redundancy or synergy in a representation (i.e., the posterior distribution) through a scalar parameter $\alpha$ to see how different metrics react to the changing representation.
> >
> > Whether redundant or synergistic entanglement arises in learned representations is analyzed in the empirical analyses in Sec.5.2.
> >
> >
> > > I am concerned about the practicality of the proposed metric as it is computed by summing over many small probability density values...
> >
> > The summation over the dataset is numerically stabilized by logsumexp function of PyTorch as $\log \sum_x p(\mathbf z_S|x) = \textbf{logsumexp}_x (\log p(\mathbf z_S|x))$, with which we can avoid numerical instability caused by tiny inputs. We found this point non-trivial, and so added a mention to Sec.3.5. Thank you for catching this point!
> >
> > As for the dimensionality, we used six latent variables in all the models as stated in the second paragraph of Sec.5.2. It does not mean the final feature dimensionality in Table 2 is six; we updated the caption of Table 2 to elaborate this point. Thank you again for the comments; we believe it improves the reproducibility aspect of our work!
> >
> >
> > > Clipping $I(y_k, z_\ell) - I(y_k, z_{\setminus\ell})$ to be >= 0 makes the metric unable to compare between models that have $I(y_k; z_\ell) \ge I(y_k; z_{\setminus\ell})$.
> >
> > We clipped the values because we understand these terms as lower bounds of PID terms, and negative bounds are vacuous. We used the results to analyze PID terms through bounds rather than to analyze bounds themselves. In some cases, we may give meaning to bounds themselves, as we did for the lower bound of redundant information; it may be meaningful to compare the values before clipping in such a case. We did not do such analysis in the paper as that is not our main motivation.
> >
> >
> > > In Fig. 3a, UniBound produces very different results for FactorVAE on dSprite and on 3DShapes. I would like to hear the explanation for this.
> >
> > We have no clear answer for this question. We understand that dSprites and 3dshapes have different aspects in terms of how each factor affects the rendering of images. For example, every factor affects the same pixel region of the image in dSprites, while in 3dshapes, some factors affect different regions of the image like floor, wall, and object. In that sense, we expect that the same model may behave differently on disentangling factors in these datasets.

---

### Official Review · Reviewer_mDLP · 2021-11-03

**Correctness:** 4
**Technical Novelty And Significance:** 3
**Empirical Novelty And Significance:** 2
**Recommendation:** 6
**Confidence:** 3

**Main Review:**

Originality:
Analyzing latent space and quantifying disentanglement is an open question in the representation learning field. The authors used PID to generalize the concept of information gap for multivariate analysis to assess the level of entanglement and disentanglement, which is rather novel and quite valuable. Deriving upper and lower bounds for the introduced partial information terms as a function of interaction information, $\mathcal{I}(.)$, is original. The entanglement attacks design is not new but is well-justified for this work.

Strengths:
- The manuscript is well organized and well written.
- The paper attempts to make a connection between the existing literatures in information gap theory and partial information decomposition to improve the entanglement analysis.
- The experimental studies illustrate that the proposed metric successfully reveals the disentanglement decrement under entanglement attacks, while the comparable measures fail.


Limitations:
- Ablation study is a critical study for this topic. I expected to see detailed ablation studies, beyond the suggested entanglement attacks.
- There is no analysis for robustness of partial information measures (the proposed bounds).
- I think the traversal analysis is missing in this work, especially in entanglement attacks. It would be quite informative if the authors report some visual results.
- I think the recent work by Do and Tran, ICLR2020, which formulates the disentanglement as another set of information-theoretic metrics is quite relevant to this work. It would be informative if the authors elaborate on those metrics as well and show which dimension(s) of information is missing in each framework.

Additional Comment/Questions:
- For each VAE model, how does sampling affect the disentanglement metrics? Was the sensitivity of partial information calculation to the number of samples studied?
- I wonder about entanglement attacks that may withstand being revealed by the proposed partial information measures. Any thoughts?
- If I understand correctly, for JointVAE, the upper and lower bounds of partial information are quite close to each other (the bound is very tight). What does this exactly mean? Why does it happen mainly for JointVAE?
- Did you consider both continuous and discrete variables when reporting the disentanglement?
- Batch size for TCVAE is quite large. Is this the case in the original implementation as well?
- The qualitative summary in Table 1 can be reported in the appendix and instead the bounds for each partial information can be reported. Or at least those qualitative measures can be elaborated in the main text.


**Summary Of The Paper:**

The paper proposes an information theoretic approach to analyze the disentanglement. Using Partial Information Decomposition (PID) framework, the authors attempt to characterize the interaction among latent factors, correlation/disentanglement, for more than two latent variables. They generalize the existing mutual information gap (MIG) metric for multiple latent variables, by decomposing mutual information term into unique, redundant, and complementary information. The authors discussed that the pairwise correlation analysis endorses a portion of the redundant information as a measure of disentanglement, which should have been considered as entanglement. Additionally, they derive bounds for the partial information terms and demonstrate the limitations of existing disentanglement measures, i.e., BetaVAE, FactorVAE, and MIG in quantifying multivariate correlation in the latent space. The authors also designed two entanglement attacks to assess the disentanglement measures under injected correlation to the latent representation. The authors used two datasets, dSprites and 3dshapes to empirically substantiate their findings.

**Summary Of The Review:**

I think this is a good submission and the proposed information-theoretic framework is persuasive. However, I am on the borderline between acceptance and rejection. The main contribution seems limited and the ablation study including entanglement attacks only considers a few simple cases of entanglements.

---

> ### Author Response · Authors · 2021-11-19
> **Response to Reviewer mDLP (1/2)**
>
> Thank you for the review. We appreciate your thorough considerations on improving the work.
>
>
> > There is no analysis for robustness of partial information measures (the proposed bounds).
>
> We considered robustness against two kinds of randomness: (1) evaluation time randomness and (2) training time randomness. For (1), we tried multiple seeds to compute the scores. See Fig.10-11 for the results; we observed UniBound is very stable against randomness in evaluation. For (2), we tried multiple seeds to train each model; most figures except the above ones and the evaluation with entanglement attacks include this randomness. We observe larger deviations than (1), which are still small enough to make qualitative statements on the results.
>
> We also conducted additional experiments with varying hyperparameters in Appendix G. The results show that the proposed bounds consistently behave when regularization hyperparameters change.
>
>
> > I think the traversal analysis is missing in this work, especially in entanglement attacks. It would be quite informative if the authors report some visual results.
>
> We do not analyze latent traversal because it is not suitable for capturing multi-variable interactions. In latent traversal, one varies only one latent variable at a time. It is a fundamental limitation of this visualization method, as we cannot place images at various points in multidimensional latent space to a two-dimensional figure. Since we are interested in how multiple variables are related, this limitation is critical for this work.
>
> Multidimensional entanglement may not be found through visualization. We argue that understanding of such a multidimensional phenomenon should be driven by quantitative measurements rather than qualitative analysis of visualization. We implemented such quantitative measurements through this work and found the existence of multidimensional entanglement through experimental results (e.g., the difference between MIG and UniBound in FactorVAE and JointVAE is a signal of the existence).
>
>
> > ...the recent work by Do and Tran, ICLR2020, which formulates the disentanglement as another set of information-theoretic metrics is quite relevant to this work.
>
> Thank you for pointing it out. We have updated the paper to include references to their work and elaborated in Appendix B. We also left some comments on some of the metrics below.
>
> WSEPIN has a similar interpretation as our framework: it can be seen as an upper bound of the unique information $\mathcal U(x; z_\ell \setminus \mathbf z_{\setminus\ell})$. It is still different from our metric in many aspects, e.g., (1) it is an optimistic evaluation in the PID perspective as it upper bounds the uniqueness while our UniBound is a pessimistic evaluation, and (2) it targets the information of $x$ in general rather than individual generative factors, with which we cannot analyze the information distribution for each factor as we did in Figure 4 and Table 5.
>
> RMIG extends MIG to imbalanced datasets. We think its main technique of inverting the conditioning of $y_k$ is also applicable to our framework. We conducted experiments on balanced datasets because whether the factors are balanced or not is not our main interest here, and hence used the original MIG.
>
> For JEMMIG, we are not very sure how its definition relates to our framework. It penalizes $H(z_\ell|y_k)$ more than (R)MIG, though this term already affects the score negatively in (R)MIG. Please see Appendix B for other metrics in the literature.
>
>
> > For each VAE model, how does sampling affect the disentanglement metrics? Was the sensitivity of partial information calculation to the number of samples studied?
>
> We illustrated the effect of sampling on evaluation in Appendix I, where we ran evaluation four times with different random seeds. We observed significantly smaller noise due to sampling compared to the training randomness. We did not empirically analyze the effect of sample size, but we expect that the score variance scales inverse proportionally to the sample size as the score is a simple Monte-Carlo estimation.
>
> Note that we used the sample size of 10,000, which comes from the choice made by the MIG paper (Chen et al, 2018). We added a note on it to Sec.3.5 (thank you for reminding us!).
>
>
> > I wonder about entanglement attacks that may withstand being revealed by the proposed partial information measures.
>
> We are also interested in such attacks if exist :)  Anyway, finding such examples should involve a new perspective and a new definition of disentanglement, which we believe promotes our understanding on this topic.

---

> > ### Author Response · Authors · 2021-11-19
> > **Response to Reviewer mDLP (2/2)**
> >
> > > If I understand correctly, for JointVAE, the upper and lower bounds of partial information are quite close to each other (the bound is very tight). What does this exactly mean? Why does it happen mainly for JointVAE?
> >
> > When $I(y_k; \mathbf z_{\setminus \ell}) \approx I(y_k; \mathbf z)$, the upper bound of $\mathcal U(y_k; z_\ell \setminus \mathbf z_{\setminus \ell})$ is close to zero and the redundancy becomes close to $I(y_k; z_\ell)$. The converse holds, too. In that sense, JointVAE provides a representation in which removing any one variable does not affect what information is represented about each factor.
> >
> > We do not have a clear explanation for why it happens only in JointVAE, though the newly added results in Appendix G on the influence of regularization hyperparameters in each method may provide some insights on understanding the phenomenon. We evaluated the JointVAE model with the varying hyperparameter $C_z$ (the final capacity of continuous variables) and found that enforcing small capacity recovers positive unique and complementary information. This result indicates that the capacity parameter plays a key role in vanishing unique information.
> >
> >
> > > The qualitative summary in Table 1 can be reported in the appendix and instead the bounds for each partial information can be reported.
> >
> > Thank you for the suggestion. We have updated the paper to add the quantitative results to the main text (Figure 4) and moved the qualitative summary to the appendix (Table 5). We also left the large version of the plots in the appendix as before.
> >
> >
> > > Did you consider both continuous and discrete variables when reporting the disentanglement?
> >
> > We considered both continuous and discrete variables on computing disentanglement scores.
> >
> >
> > > Batch size for TCVAE is quite large. Is this the case in the original implementation as well?
> >
> > Yes, we selected it following the original work. We understand that this is required to make total correlation estimation stable.
> >
> >
> > > I expected to see detailed ablation studies…
> >
> > As we noted above, we added evaluation results for models trained with varying regularization hyperparameters in Appendix G. In summary, stronger regularization resulted in larger unique information. In JointVAE, we observed that the high redundancy is caused by the large capacity of continuous variables derived from the original paper (Dupont, 2018); with smaller capacity, the redundancy is reduced and the unique information is recovered. These results may help us to understand how our method works for analyzing disentanglement learning methods in detail.
> >
> > In terms of ablation studies of our framework itself, we have only one hyperparameter, the sample size, for which we wrote a comment above.

---

> > > ### Comment · Reviewer_mDLP · 2021-11-30
> > > **Response to the rebuttal**
> > >
> > > I appreciate authors' effort in clarifying my concerns. Some of them are more clear to me now, but some are not.
> > >
> > > - **robustness against two kinds of randomness:** Looking at the results in Figures 10 and 11, it seems the proposed bound has high variance and is sensitive to the random initialization.
> > >
> > > - **Multidimensional entanglement may not be found through visualization:** I agree that multidimensional entanglement is more complicated than one dimensional case. However, for simple datasets such as dSprites that are generated to study disentangled representations, I think it is quite useful to provide a qualitative assessment, which helps to better interpret the entanglement measures.
> > >
> > > - **Do and Tran, ICLR2020:** I appreciate authors' effort in addressing this paper, but it would be more informative to report the values of those measures, e.g., WSEPIN, in your figures.
> > >
> > > Overall, I think the authors did put a lot of effort into this work. I would like to keep my score slightly above the border line, but not higher.

---

> > > > ### Author Response · Authors · 2021-11-30
> > > > **Re: Response to the rebuttal**
> > > >
> > > > Thank you for the response. We would like to comment on **robustness** against training randomness. We emphasize again that the deviation caused by training time randomness is small enough to compare models (Fig.3 and 4) and hyperparameter choices (Fig.7). Note that the variance of UniBound is not consistently larger nor smaller than MIG, and the difference is small if not zero.

---

### Official Review · Reviewer_25MQ · 2021-11-04

**Correctness:** 4
**Technical Novelty And Significance:** 4
**Empirical Novelty And Significance:** 3
**Recommendation:** 8
**Confidence:** 4

**Main Review:**


Good:

The paper is well written, the explanations are very clear. Not only the theory, but the experiments as well, e.g. Table 1 captures the high level experimental results nicely that otherwise would have been hard to decipher from the raw data (in the appendix)

The ideas behind the metric are intuitively appealing. The 3 main information terms are well defined and explained. The changes to MIG metric makes sense in this light, as it better captures which term should be counted positively and negatively in the metric (Fig 2)

The paper shows 2 adversarial attacks, where the latent vectors are perturbed with noise so each attack targets the redundancy and synergy term respectively.
- It is shown theoretically on a toy example that UniBound responds better to redundancy attacks, i.e. converges to zero faster with more noise. For synergy attacks they are the same
- It is shown on 2 datasets that UniBound performs better vs. other metricks against redundancy attacks.

Several disentangling methods were compared not only with the proposed metric, but analysed w.r.t the a detailed decomposition of terms.

Bad:

"and the inference distribution p(z|x) and its marginals p(zl |x), p(z\l |x) are all tractable (e.g., mean field variational models)"
Perhaps this is a strong assumption.

Minor:
The notation could be a bit better.
- The mutual information has a different font than than U, R, C. They should have the same font, as they are just a a renaming of the same function, but applied on different variables.
- The above is confusing, when the 'interaction information of a triple' I is introduced with the same font as R, R and C.


**Summary Of The Paper:**

The paper proposes a novel disentanglement metric 'Unibound'. To construct the metric the method uses partial information decomposition, which decomposes the the mutual information between the latent variables and the labels into a sum of other information terms with specific roles:
- Redundant information
- Unique information
- Complementary information

Each of these terms are precisely defined and an intuitive explanation os given.
The proposed UnibBund metric is defined similarly to MIG based on the above terms. The main difference is that UnibBund uses a better composition than MIG, that:
- better captures our intuitive notion of disentanglement
- handles adversarial representation attacks better

**Summary Of The Review:**

The proposed metric in the paper is well designed and supported by:
- intuition
- theory
- experiments vs. adversarial attacks and compared with other metrics

--------- UPDATE ---------

I have read the other reviews and rebuttals. The authors clarified some details and notations. I keep my score and recommend the paper for publication.

---

> ### Author Response · Authors · 2021-11-19
> **Response to Reviewer 25MQ**
>
> Thank you for the review. We highly appreciate that you positively evaluate our work!
>
>
> > "and the inference distribution p(z|x) and its marginals p(zl |x), p(z\l |x) are all tractable (e.g., mean field variational models)" Perhaps this is a strong assumption.
>
> We may opt to quantization-based computation for a modest number of latent variables (e.g., six as we used in the experiments).
>
>
> > The mutual information has a different font than U, R, C. They should have the same font, as they are just a renaming of the same function, but applied on different variables.
>
> We used different fonts between mutual information and PID terms to emphasize that PID terms are NOT mutual information themselves; i.e., they are not just renamings of the same function! They express the information distribution among more than two variables (namely $y_k$, $z_\ell$, and $\mathbf z_{\setminus\ell}$ in our case), while mutual information expresses the information between two variables. The information distribution among three variables are related with (but not fully determined by) mutual information through Eq.1-2. They are also illustrated in Fig.1 and Fig.2a.
>
>
> > The above is confusing, when the 'interaction information of a triple' I is introduced with the same font as R, R and C
>
> We have changed the notation to avoid confusion. Thank you for pointing it out!

---

### Author Response · Authors · 2021-11-19
**To All Reviewers**

We would like to thank all the reviewers for detailed and constructive comments. We have added replies to each thread. We have also revised the paper in response to the comments; in particular, we have

- replaced the table of qualitative interpretations for factor-wise PID analyses with quantitative results (Figure 4 is the plots added to the main text, and the table is moved to Appendix E with additional notes that the interpretations may be affected by experimental settings);
- added citations and discussions on other metrics as suggested by some reviewers (see Sec.2, Sec.3.2, and Appendix B);
- added experiments on varying regularization hyperparameters in Appendix G; and
- added measurements of total correlation for JointVAE in Appendix F (Table 6).

---

### Decision · Program_Chairs · 2022-01-20

**Decision:**

Accept (Poster)

**Comment:**

The topic of the paper is the use of partial information decomposition (PID) for the analysis of interactions in latent representations.

All reviewers ended up appreciating the paper after a good extensive discussion with the authors. The numerical investigation is somewhat on the short side. One reviewer asks for more ablation studies and one reviewer asks for more investigation on real datasets to show the advantage of the method.

The paper is borderline. The theoretical development is fine. But one could argue that the paper could benefit from some more work on the experiments. However, the main points of the method is in place and further validation of the method can be left for future contributions.